# Debate as Reward: A Multi-Agent Reward System for Scientific Ideation via RL Post-Training

## Abstract

Large Language Models (LLMs) have demonstrated potential in automating scientific ideation, yet current approaches relying on iterative prompting or complex multi-agent architectures often suffer from hallucination or computational inefficiency. A critical bottleneck in applying Reinforcement Learning (RL) to this open-ended domain is reward hacking—where models exploit imperfect evaluation proxies to maximize scores without producing genuine scientific innovation. To address these limitations, we propose an RL framework explicitly tailored for high-quality scientific idea generation. We propose the first multi-agent reward function designed to serve as a judge, decoupling methodological validation from implementation details while providing strict binary rewards that are robust to reward hacking. To effectively optimize against this sparse signal, we utilize an unbiased variant of Group Relative Policy Optimization to mitigate artificial length bias. We ground our training in a curated dataset of problem-solution pairs extracted from ICLR 2024 and NeurIPS 2025 proceedings. Experiments demonstrate that our framework significantly outperforms state-of-the-art baselines across expert-evaluated metrics of novelty, feasibility, and effectiveness. Our code is available at `https://anonymous.4open.science/r/debate-as-reward-80AF`.

## 1 Introduction

Scientific progress fundamentally depends on the generation of novel, feasible, and impactful research ideas. While large language models (LLMs) have shown promise in accelerating scientific discovery—assisting in hypothesis formulation, experimental design, and literature synthesis—their ability to produce genuinely innovative ideas remains limited Shahhosseini et al. (2025). Current approaches predominantly rely on inference-time augmentation strategies.

Our work specifically targets the generation of a high-quality scientific idea that directly addresses a user-provided research question. Although the existing methods demonstrate utility, they operate atop frozen base models and fail to internalize the sophisticated, multi-step reasoning required to bridge open-ended research questions with methodologically sound and conceptually novel solutions. Consequently, their generated ideas often lack depth, technical rigor, or true novelty when subjected to expert or systematic evaluation Guo et al. (2024); Si et al. (2024); Wang et al. (2024b).

In parallel, recent advances in reinforcement learning (RL) post-training—particularly Group Relative Policy Optimization (GRPO) Shao et al. (2024) and Proximal Policy Optimization (PPO) Schulman et al. (2017)—have significantly enhanced the performance of large language models (LLMs) on reasoning-intensive tasks, such as mathematical problem solving Wang et al. (2025) and code generation Robeyns & Aitchison (2025). Building on this foundation, researchers have begun exploring RL-based alignment for higher-level cognitive tasks beyond formal reasoning. Notably, Li et al. (2024) demonstrated that RL post-training can also improve scientific idea generation by aligning model outputs with human judgments of novelty and feasibility—suggesting that the benefits of policy optimization extend from structured reasoning to open-ended, creative scientific synthesis.

A central challenge remains: designing a reward signal that is both accurate—capturing nuanced dimensions such as novelty and feasibility—and robust against exploitation, or "reward hacking," where models learn to

game superficial markers of quality without producing substantive innovation Skalse et al. (2022). To address this, we developed and evaluated over fifty distinct prompting strategies for our judge LLM. Through iterative qualitative analysis of model outputs during training, we identified numerous forms of reward hacking. Some were implicit and common, such as generating excessively long ideas, becoming overly specific yet irrelevant or remaining overly vague and generic. More strikingly, we observed a sophisticated failure mode where the generator model intentionally outputs incomplete placeholder phrases, such as "... the answer part ..." (see Appendix H for details). Because the judge model is provided with both the research question and the ground-truth abstract as context, these placeholders inadvertently act as completion prompts. The judge is thereby tricked into generating the missing scientific idea itself within its reasoning section. It then compares its own generated idea against the abstract—naturally finding a strong methodological match—and erroneously assigns a reward of 1. Thus, the generator successfully exploits the judge's generative capabilities to bypass the ideation process entirely.

Compounding this issue, the evaluation of scientific ideas remains inherently difficult. Many studies resort to "LLM-as-a-judge" paradigms Baek et al. (2025); Su et al. (2025); Li et al. (2025) or embedding-based similarity metrics Keya et al. (2025); Wang et al. (2024a) to score outputs along dimensions such as novelty, feasibility, and effectiveness. Yet LLMs struggle to reliably assess abstract qualities like conceptual novelty, and encoder-based metrics often conflate semantic similarity with true innovation.

We address these challenges through a novel RL post-training framework that aims to approximate aspects of scientific reasoning directly into a compact LLM. Our core insight is that high-quality idea generation is best achieved not by adding complexity at inference time, but by fundamentally reshaping the model's generative policy through targeted training on high-impact scientific exemplars. To construct a robust reward signal, we leverage the observation that LLMs are more reliable at detecting similarity than novelty Zheng et al. (2023). Accordingly, we curate a set of high-quality, accepted papers from ICLR 2024 and NeurIPS 2025 that were published after the model's knowledge cutoff date, and treat each paper's abstract as a "target" idea. We then used an LLM prompted to decide whether a generated idea matches the target abstract, and trained the model on a subset of the ICLR 2024 dataset.

To mitigate reward hacking, we systematically refined the judge's prompting strategy and incorporated multi-agent deliberation. We note that, unlike prior work that uses multi-agent LLM judges after generation Baek et al. (2025); Assafelovic (2023), we are the first to integrate multi-agent judge directly into the online RL training loop as the reward function. To quantify this refinement, we established a controlled empirical protocol using a fixed, expert-annotated validation set comprising 177 (research question, abstract, generated idea) triples collected across multiple training runs. Each sample was rigorously labeled via majority vote by seven domain experts. Every prompt modification and architectural change was assessed over five independent runs to mitigate stochastic variance.

Our initial single-call binary judge achieved a mean precision of 0.85, remaining vulnerable to verbosity bias and superficial lexical overlap. A reduced multi-agent setup utilizing only a single LLM analyst improved precision to 0.906 but still permitted false-positive rewards. Ultimately, we converged on a multi-agent deliberative judge that explicitly separates methodological decomposition and final aggregation. This final system achieves perfect precision (1.00) on the expert benchmark. Because false-positive rewards are the primary driver of reward hacking in RL settings, eliminating them is paramount to our framework's success. Full details of the prompt evolution, strategy comparisons, and comprehensive ablation studies are provided in Appendix B.

We define the input to our system as a problem-only research question: a concise, self-contained query that articulates the core scientific challenge addressed in a paper, while deliberately omitting any trace of its proposed solution or method. These questions are generated by prompting Gemini 2.5 pro to analyze a paper's title and abstract and extract only the underlying problem, following strict guidelines that prevent information leakage from the "golden idea." For example, given the research agent Baek et al. (2025) paper, the research question would be "How can the generation of novel scientific research ideas be automated and accelerated?"

Trained exclusively on ICLR 2024 papers via GRPO, our model learns to map research questions to plausible methodological proposals, demonstrating that RL post-training can effectively enhance the scientific ideation capabilities of small, efficient models.

For final evaluation, we employ two independent LLM judges (one providing absolute scores, the other pairwise comparisons) on a subset of both the ICLR 2024 and NeurIPS datasets, complemented by manual relative assessments on 10 unsolved research questions authored by domain experts.

Our contributions are threefold:

- The first GRPO-based post-training framework explicitly designed for scientific idea generation, moving beyond inference-time augmentation to internalize deep reasoning capabilities.

- We propose the first multi-agent LLM judge system deployed in an online RL post-training loop for LLMs, enabling robust, consensus-driven reward signals that mitigate reward hacking.

- Empirical validation showing significant improvements over supervised fine-tuning (SFT) and strong baselines in both automatic (LLM-based) and human expert evaluations.

## 2 Related Work

**Automated Research Idea Generation** The capability of Large Language Models to generate novel scientific ideas has attracted significant attention. Early approaches such as SciMon Wang et al. (2024a), leveraged retrieval-augmented generation to produce new ideas based on prior literature, focusing on novelty optimization. This was further advanced by ResearchAgent Baek et al. (2025), which introduced an iterative refinement process where LLMs critique and improve their own ideas using knowledge graphs and literature entities. Similarly, GPT Researcher Assafelovic (2023) leverages a plan-and-solve agentic architecture to automate web scraping, filtering and comprehensive literature synthesis, thereby streamlining research workflows.

More recently, the field has moved toward end-to-end automation. The AI Scientist Lu et al. (2024) and its successor The AI Scientist-v2 Yamada et al. (2025) propose fully automated pipelines that not only generate ideas but also execute code and write papers, using techniques such as agentic tree search. Concurrently, CycleResearcher Weng et al. (2025) improves automated research outcomes by integrating a review-refinement loop directly into the generation cycle.

Other works have focused on the structural quality of ideas. Chain of Ideas Li et al. (2025) organizes the generation process into sequential chains of literature to enhance logical flow, while LDC Li et al. (2024) introduces dynamic control mechanisms to guide the generation path. Similarly SPARK Sanyal et al. (2025) emphasizes creative generation through system-level architecture. However despite these advances, most existing methods rely on inference-time prompting strategies or extensive retrieval pipelines with frozen models. In contrast our approach focuses on internalizing the reasoning capabilities required for idea generation by fine-tuning a smaller and efficient model via reinforcement learning.

**LLM-based Evaluation and Multi-Agent Debate** Evaluating open-ended text generation, particularly in scientific domains, remains a critical bottleneck. IdeaBench Guo et al. (2024) and recent large-scale human studies Si et al. (2024) highlight the difficulty of assessing novelty and feasibility without human experts. To mitigate this, recent works have turned to agentic frameworks. Generative Agent Reviewers Bougie & Watanabe (2025) simulates the peer review process.

Multi-agent debate has emerged as a powerful paradigm for improving accuracy. ChatEval Chan et al. (2024) demonstrates that a committee of agents discussing open-ended questions can converge on higher-quality evaluations than single agents. We adopt this insight not merely for final evaluation, but as a richer, more reliable reward signal. By employing a multi-agent debate system (simulating diverse roles to judge methodological alignment with ground truth), we construct a robust reward function that guides the training of our smaller model.

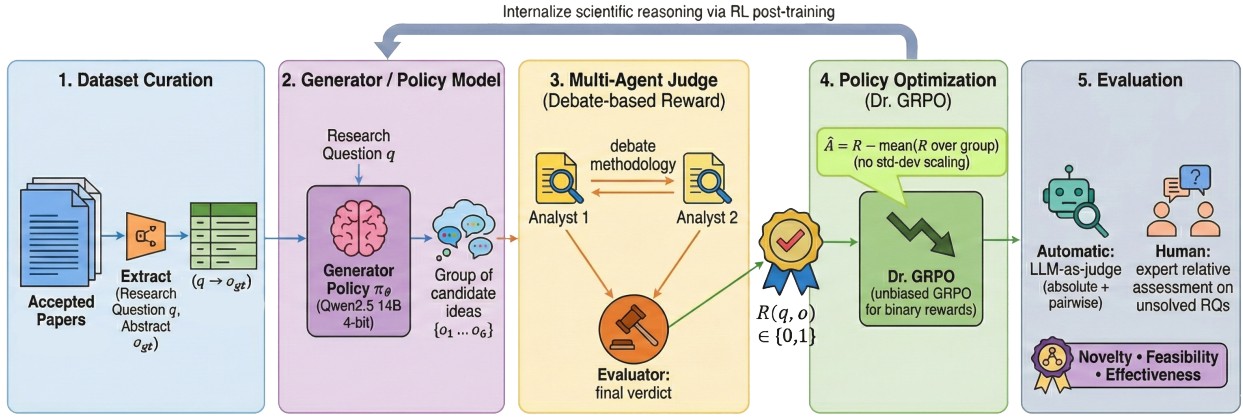

Figure 1: **Overview of the proposed framework for internalizing scientific reasoning via Reinforcement Learning (RL) post-training.** The pipeline consists of five distinct stages: (1) **Dataset Curation**, where research questions and abstracts are extracted from accepted ICLR 2024 and NeurIPS 2025 papers to form the training dataset; (2) **Candidate Generation**, utilizing a Qwen2.5 14B policy model to propose research ideas; (3) **Multi-Agent Judge**, employing a debate-based mechanism (two Analysts, and one Evaluator) to assign binary rewards; (4) **Policy Optimization**, utilizing Dr. GRPO (Group Relative Policy Optimization) to update the model without standard deviation scaling; and (5) **Final Evaluation**, assessing the trained model for novelty, feasibility, and effectiveness using both LLM-as-a-judge and human expert review.

**Reinforcement Learning and Self-Improvement**  While Supervised Fine-Tuning (SFT) aligns models with instruction formats, Reinforcement Learning (RL) encourages complex reasoning. Techniques like Self-Rewarding Language Models Yuan et al. (2024) allow models to generate their own training signals. In the mathematical domain, DeepSeekMath Shao et al. (2024) introduced Group Relative Policy Optimization (GRPO), a memory-efficient variant of PPO Schulman et al. (2017) that eliminates the need for a value function critic by normalizing scores within a group of outputs.

Our work bridges the gap between structured mathematical reasoning and open-ended scientific discovery by adapting the GRPO framework to the task of research idea generation. This method enables the model to internalize the subtle reasoning patterns required to bridge an open research question to a methodologically sound and novel solution.

## 3 Methodology

We propose a closed-loop framework designed to automate the generation of high-quality, novel scientific research ideas. Our approach integrates a specialized research dataset, a deliberative multi-agent reward model, and a recent reinforcement learning policy optimization strategy. Formally, the pipeline consists of three phases: (1) Dataset Curation, where we extract problem-solution pairs from high-impact literature; (2) Multi-Agent Evaluation, where a committee of LLM agents provides robust feedback; and (3) Policy Optimization, where the generator is fine-tuned via an unbiased version of Generative Reinforcement Policy Optimization (Dr. GRPO).

### 3.1 Overview of the Framework

We conceptualize the research idea generation task as a conditional sequence generation problem. Let $\pi_\theta$ denote the generator policy parameterized by $\theta$. Given a research question **q**, the model generates a candidate solution **o**. To optimize $\pi_\theta$, we employ a Reinforcement learning paradigm (RL) where the reward signal is not derived from static ground truth but is synthesized by a Multi-Agent Judge System.

As illustrated in Fig. 1, the training cycle proceeds as follows:

1. **Sampling:** For a given question $\mathbf{q}$, the generator samples a set of $G$ candidate ideas $\{\mathbf{o}_1, \ldots, \mathbf{o}_G\}$ (Figure 8).

2. **Deliberation:** The Multi-Agent Judge evaluates each candidate $\mathbf{o}_i$ by comparing it against the ground-truth abstract $\mathbf{o}_{\mathrm{gt}}$ and the original question $\mathbf{q}$, both extracted from the target paper. Through an adversarial dialogue among multiple LLM-based agents, the system produces a binary reward $R(\mathbf{q}, \mathbf{o}_i; \mathbf{o}_{\mathrm{gt}})$ indicating whether $\mathbf{o}_i$ meaningfully aligns with the core contribution of $\mathbf{o}_{\mathrm{gt}}$.

3. **Optimization:** We update $\theta$ to maximize the expected reward using Dr. GRPO, estimating advantages based on the group outputs.

### 3.2 Multi-Agent Evaluation Framework (The Judge)

A critical bottleneck in RLHF (Reinforcement Learning from Human Feedback) for open-ended tasks is the reliability of the reward model. Single-model evaluators are prone to "reward hacking" where the generator optimizes for length or keyword frequency rather than quality. To mitigate this, we introduce a Deliberative Multi-Agent Judge, simulating a peer-review committee.

We instantiate three distinct agents using the GPT-OSS 120B model. The evaluation protocol is structured as a hierarchical dialog.

The framework enforces a strict evaluation criterion: agents must focus exclusively on methodology, novelty, and critical assumptions, explicitly ignoring non-conceptual details such as dataset choices, evaluation metrics, or experimental setups.

1. **The Analyst (Methodological Investigator):** This agent's core responsibility is to decompose the ground-truth Idea into its fundamental methodological elements (Figure 16). It decomposes the generated idea $\mathbf{o}$ into technical components. It compares $\mathbf{o}$ against the ground truth $\mathbf{o}_{gt}$ to identify aligned contributions and potential hallucinations, presenting a structured feasibility report (e.g., specific algorithms, architectural innovations, or theoretical assumptions).

2. **The Evaluator (Final Decision Maker):** The Evaluator functions as the meta-reviewer and final aggregator (Figure 20). It consumes the full interaction history between two Analysts. By weighing the technical merits against the exposed flaws, it synthesizes a final binary reward ($r \in \mathbb{R}$) alongside a structured reasoning trail.

Appendix C provides the detailed operational workflow of the broader multi-agent architecture search. To make this interaction concrete, we provide a condensed example of an actual multi-agent judgment in Figure 28. In this example, one Analyst initially leans toward assigning a negative reward because the generated idea omits several details from the target abstract. However, after considering the other Analyst's argument in the following round, it revises its assessment by distinguishing the core methodological contribution from secondary implementation or application details. The Evaluator then aggregates the debate history and assigns the final binary reward. This example illustrates how the framework goes beyond a static similarity check: the Analysts first decompose and challenge the methodological match, and the Evaluator converts the resulting discussion into a calibrated reward signal.

Empirically, this multi-turn adversarial dynamic improves reward reliability by reducing false-positive rewards. In our reward-model validation benchmark, static and non-deliberative alternatives produced substantially more false-positive rewards than the final multi-agent judge: for example, the single-call binary judge achieved 0.85 precision, corresponding to 15% false positives among positively rewarded samples, whereas the final multi-agent judge achieved 1.00 precision on the same benchmark. As reported in Appendix B and Table 8, this trend holds across several static alternatives, including embedding similarity, structured compliance, keyword heuristics, and other non-debate evaluation strategies. The ablation results in Section 4.5 further show that removing key debate components, such as the Analyst or Evaluator, substantially changes the precision-recall behavior of the reward model. These results support the claim

that the debating framework improves reward verification by transforming evaluation from a static scoring task into a structured process of methodological decomposition and final aggregation. Through adversarial cross-examination, agents are forced to defend their assessments, exposing logical gaps and hallucinated complexity that evade single-pass detection. This helps ensure that rewards target the intrinsic logic of the scientific mechanism rather than mere lexical overlap or specific implementation contexts.

### 3.3 Policy Optimization via Dr. GRPO

We fine-tune the generator $\pi_\theta$ using Dr. GRPO Liu et al. (2025), an unbiased variant of Group Relative Policy Optimization. Unlike PPO, GRPO estimates baselines directly from group scores of sampled outputs, eliminating the need for a separate value network and reducing memory overhead—critical for our 14B parameter model.

However, as showd by **??** standard GRPO suffers from a length bias: it applies sequence-level advantages uniformly across tokens, causing longer outputs to accumulate disproportionately large gradients regardless of quality. This is particularly problematic for scientific ideation, where verbose but vacuous responses can exploit the reward signal. Dr. GRPO addresses this by introducing length-normalized token-level advantages:

$$\hat{A}_{i,t} = \hat{A}_i \cdot \frac{|\mathbf{o}_i|^{-1}}{\frac{1}{G}\sum_{j=1}^{G} |\mathbf{o}_j|^{-1}} \tag{1}$$

where $\hat{A}_i$ is the standardized sequence-level advantage. This normalization ensures each response contributes equally to the gradient irrespective of length, improving token efficiency and preventing the model from favoring padded elaborations over concise, high-quality ideas.

The Dr. GRPO loss follows the standard clipped objective:

$$\mathcal{L}_{\text{Dr.GRPO}}(\theta) = \frac{1}{G}\sum_{i=1}^{G}\sum_{t=1}^{|\mathbf{o}_i|} \min\left[\rho_{i,t}(\theta)\hat{A}_{i,t}, \text{clip}\left(\rho_{i,t}(\theta),\, 1-\epsilon,\, 1+\epsilon\right)\hat{A}_{i,t}\right] \tag{2}$$

where $\rho_{i,t}(\theta)$ is the importance sampling ratio at token $t$, and $\epsilon$ is the clipping hyperparameter. This formulation, combined with our binary reward signal from the Multi-Agent Judge, provides strong gradients that reinforce methodologically sound ideas without the length exploitation common in open-ended generation tasks.

## 4 Experiments and Results

### 4.1 Implementations

We implement our method and baselines in two configurations: (1) using the Unsloth-optimized, 4-bit quantized version of Qwen 2.5 14B for computationally efficient deployment, and (2) using the full-precision Qwen 2.5 7B model to assess performance without quantization artifacts (results reported in Section 4.5.4).

Many baseline methods are inherently expensive during inference; agentic systems (e.g., GPT Researcher, ResearchAgent) involve numerous internal LLM calls and iterative refinement loops, while other methods utilize costly per-token decoding schemes. In contrast, our model performs a single, highly efficient forward pass. To ensure a fair, compute-matched evaluation against these resource-intensive baselines, we evaluate our proposed method using a Best-of-10 (BoN(10)) approach. For each research question, we generate 10 candidate ideas and prompt the same model to select the one it deems most novel (see Figure 10 for the Best-of-N prompt). This strategy aligns our inference budget with the more complex methods, ensuring an equitable comparison of final output quality for a similar amount of compute without requiring external evaluators.

To account for randomness during training and judgment, we train our method four independent times. For Table 2, each checkpoint is evaluated once—giving four judge runs total—and all baselines are re-evaluated in every run to account for evaluator variance.

## 4.2 Data

To train and evaluate our model, we collected accepted papers from ICLR 2024 and NeurIPS 2025, then removed any papers archived before model's knowledge cutoff date. Next, we plotted score histograms for each set and retained about the top 50% of papers with the highest scores. Since our task specifically required papers containing novel ideas, we used the GPT-OSS-120B model to classify the papers into three categories: surveys, evaluation-focused papers, and those presenting novel ideas (see Figure 4 for the prompt details). We retained only the latter category for our dataset.

Next, we extracted the full text of each paper, spanning from the abstract up to (but not including) the references section. We then passed this full text to DeepSeek-V3.1 to extract the "golden idea" of each paper (see Figure 22 for the Idea Extraction prompt). For ICLR papers, we extracted the research question using Gemini 2.5 Pro, while for NeurIPS papers, we used DeepSeek-V3.1 with the same input prompt (as shown in Figure 6), supplemented by the abstract of each paper. Initially, we relied solely on abstracts under the assumption that the central research question would be clearly stated there; however, the resulting extracted ideas lacked sufficient specificity. Consequently, we switched to using the full text of the papers to extract more precise and meaningful golden ideas.

During dataset creation, whenever we used LLMs for any purpose, we carefully analyzed their outputs and either corrected the results when necessary or revised the input prompts if a significant number of records exhibited low quality.

Finally, we reserved 40 papers from ICLR 2024 and 40 papers from NeurIPS 2025 as the RL test sets for evaluation. For supervised fine-tuning (SFT), we used a subset of the ICLR 2024 data for training and the remainder for validation (80 and 16 samples, respectively); no SFT data was used from NeurIPS 2025. For RL post-training, we used the ICLR 2024 training split (320 samples) and the NeurIPS 2025 training split (2,156 samples).

Table 1: Dataset splits for RL and SFT

| Dataset | ICLR 2024 | NeurIPS 2025 |
|---|---|---|
| RL Train | 320 | 2156 |
| RL Validation | 40 | 270 |
| RL Test | 40 | 270 |
| SFT Train | 80 | - |
| SFT Validation | 16 | - |

In addition to these conference papers, we incorporated two supplementary sources: (1) 203 AI-generated papers from AI Scientist Lu et al. (2024), from which we extracted research questions using their abstracts using DeepSeek-V3.1; (2) A curated set of 30 active research questions contributed by PhD candidates, each representing core components of their ongoing or future research in their respective subfields; we call these the golden RQs. We have included representative samples in Appendix F to provide insight into the nature and complexity of these questions.

## 4.3 Baselines

We compare the ideas generated from our model by greedy sampling using temperature equal to 1.0 against several common baselines in idea generation. Here are the baselines we used:

- **Unsloth Qwen 2.5 14B Instruct 4-bit**: The pretrained model quantized to 4-bit, provided with the same input prompt as our trained model to serve as a zero-shot baseline (see Figure 8 for the idea generation prompt).

Table 2: Performance comparison across different datasets for unsloth-Qwen2.5-14B 4-bit quantized. Values are reported as mean ± standard deviation across four independent training runs. Best scores are bolded.

| Dataset | Metric | Base Model 14B 4-bit | Our Method (w/o BoN) | Our Method - BoN(10) | SFT | GPT Researcher | Research Agent | AI Scientist V2 | AI Scientist | LDC |
|---|---|---|---|---|---|---|---|---|---|---|
| ICLR 2024 | Absolute Novelty | 3.77 ± 0.091 | 4.14 ± 0.044 | **4.28** ± 0.056 | 4.05 ± 0.078 | 4.10 ± 0.104 | 4.12 ± 0.117 | 3.70 ± 0.098 | - | 3.89 ± 0.11 |
| | Absolute Feasibility | 4.02 ± 0.058 | 3.82 ± 0.052 | 3.92 ± 0.056 | 3.84 ± 0.072 | 3.79 ± 0.111 | 3.99 ± 0.124 | **4.24** ± 0.078 | - | 4.02 ± 0.16 |
| | Absolute Effectiveness | 4.16 ± 0.085 | 4.46 ± 0.036 | 4.55 ± 0.044 | 4.39 ± 0.072 | 4.34 ± 0.098 | **4.69** ± 0.124 | 4.26 ± 0.091 | - | 4.11 ± 0.10 |
| | Pairwise Novelty | 1.15 ± 0.039 | 4.26 ± 0.064 | **5.00** ± 0.000 | 2.55 ± 0.124 | 4.05 ± 0.143 | 1.74 ± 0.117 | 1.05 ± 0.026 | - | 3.32 ± 0.09 |
| | Pairwise Feasibility | **5.00** ± 0.000 | 1.39 ± 0.082 | 1.68 ± 0.072 | 2.80 ± 0.117 | 2.01 ± 0.137 | 2.20 ± 0.150 | 4.35 ± 0.104 | - | 1.10 ± 0.26 |
| | Pairwise Effectiveness | 2.55 ± 0.111 | 4.91 ± 0.078 | **5.00** ± 0.000 | 1.25 ± 0.046 | 3.30 ± 0.163 | 3.90 ± 0.137 | 2.00 ± 0.117 | - | 4.42 ± 0.17 |
| NeurIPS 2025 | Absolute Novelty | 3.42 ± 0.098 | 4.06 ± 0.052 | 4.11 ± 0.064 | 3.52 ± 0.091 | 4.05 ± 0.117 | 3.96 ± 0.111 | 3.16 ± 0.091 | - | **4.28** ± 0.12 |
| | Absolute Feasibility | 3.85 ± 0.085 | 3.75 ± 0.072 | 3.90 ± 0.048 | 3.96 ± 0.078 | 3.95 ± 0.104 | 3.19 ± 0.130 | **4.12** ± 0.085 | - | 3.95 ± 0.22 |
| | Absolute Effectiveness | 3.39 ± 0.091 | 3.88 ± 0.052 | 3.99 ± 0.060 | 3.81 ± 0.085 | 4.16 ± 0.040 | 3.34 ± 0.117 | 3.81 ± 0.098 | - | **4.56** ± 0.13 |
| | Pairwise Novelty | 1.10 ± 0.033 | 4.40 ± 0.060 | **5.00** ± 0.000 | 2.10 ± 0.124 | 4.12 ± 0.137 | 2.55 ± 0.156 | 1.25 ± 0.046 | - | 3.10 ± 0.19 |
| | Pairwise Feasibility | 4.30 ± 0.040 | 1.46 ± 0.058 | 1.15 ± 0.039 | 2.91 ± 0.072 | 3.80 ± 0.111 | **5.00** ± 0.000 | 3.91 ± 0.104 | - | 2.05 ± 0.13 |
| | Pairwise Effectiveness | 1.15 ± 0.039 | 4.13 ± 0.040 | **5.00** ± 0.000 | 1.87 ± 0.117 | 4.41 ± 0.124 | 1.20 ± 0.039 | 2.15 ± 0.111 | - | 1.60 ± 0.21 |
| AI-Scientist | Absolute Novelty | 3.75 ± 0.085 | **4.18** ± 0.041 | 4.18 ± 0.044 | 3.99 ± 0.078 | 3.97 ± 0.111 | 4.05 ± 0.124 | 3.72 ± 0.091 | 3.40 ± 0.098 | 3.66 ± 0.20 |
| | Absolute Feasibility | 4.34 ± 0.072 | 4.02 ± 0.048 | 4.13 ± 0.056 | 3.47 ± 0.091 | 4.05 ± 0.098 | 3.79 ± 0.117 | 3.34 ± 0.098 | **4.38** ± 0.078 | 4.05 ± 0.05 |
| | Absolute Effectiveness | 4.05 ± 0.078 | 4.11 ± 0.044 | **4.51** ± 0.091 | 3.03 ± 0.098 | 3.99 ± 0.104 | 4.00 ± 0.111 | 4.04 ± 0.085 | 4.10 ± 0.091 | 4.08 ± 0.05 |
| | Pairwise Novelty | 2.80 ± 0.124 | 3.83 ± 0.060 | **5.00** ± 0.000 | 1.20 ± 0.039 | 3.65 ± 0.143 | 4.55 ± 0.117 | 1.70 ± 0.098 | 1.65 ± 0.091 | 2.15 ± 0.10 |
| | Pairwise Feasibility | 4.40 ± 0.091 | 1.18 ± 0.039 | 1.18 ± 0.052 | 2.70 ± 0.124 | 2.20 ± 0.143 | 2.00 ± 0.124 | 4.95 ± 0.019 | **4.98** ± 0.013 | 4.05 ± 0.14 |
| | Pairwise Effectiveness | 4.08 ± 0.104 | 4.87 ± 0.052 | **5.00** ± 0.000 | 1.60 ± 0.078 | 3.20 ± 0.060 | 2.30 ± 0.137 | 2.34 ± 0.124 | 1.10 ± 0.033 | 1.90 ± 0.06 |

- **SFT**: Supervised fine-tuning of the Qwen 2.5 14B Instruct model on our curated ICLR 2024 dataset of (research question, abstract) pairs (see Figure 12 for the SFT prompt).

- **Research Agent** Baek et al. (2025): An agentic framework that iteratively proposes, critiques, and refines research ideas using a knowledge graph and self-reflection loops. We modified its prompt to generate shorter outputs and used the method section as the idea.

- **GPT Researcher** Assafelovic (2023): A pipeline that leverages an LLM to autonomously conduct literature reviews, formulate hypotheses, and generate research proposals based on user-defined topics. We modified its prompt to generate a research idea instead of the report.

- **AI-Scientist** Lu et al. (2024): A fully automated system that generates novel scientific hypotheses, designs experiments, and writes manuscripts in specific domains.

- **AI-Scientist v2** Yamada et al. (2025): An enhanced version of AI-Scientist with improved hypothesis generation, broader domain coverage, and integrated validation modules.

Because AI-Scientist requires domain-specific code templates as input to generate papers, we evaluated all methods on a shared set of research questions extracted from existing AI-Scientist output samples. Consequently, AI-Scientist v1 was not evaluated on other datasets, as it cannot operate without its required template-based context.

## 4.4 Evaluation

**Automatic evaluation.** Following established practices in prior work, we employ two complementary LLM-based evaluation protocols: (1) *absolute scoring* (as shown in Figure 26), where each idea is independently rated on a fixed scale, and (2) *pairwise comparison* (as shown in Figure 24), where all generated ideas are evaluated against one another. We implement both to ensure robust assessment.

For evaluation, we use Qwen 2.5 72B, which shares the same knowledge cutoff date as our trained model. This ensures the evaluator is not biased by familiarity with post-cutoff papers, enabling fair and unbiased assessment of novelty and feasibility. Model selection was constrained to systems whose training data predates our dataset (September 2024) to minimize the risk that the evaluator had previously seen examples from the dataset.

In the absolute scoring setup, the evaluator assigns each idea an integer score from 1 to 5, with 5 indicating the highest quality. For pairwise comparisons, we present every unordered pair of ideas twice—once in each order—to mitigate position bias. The model's judgment for each comparison is mapped to a numerical value: +1 if the first idea is preferred, -1 if the second is preferred, and 0 if they are deemed equivalent. We aggregate these values across all comparisons for each idea and linearly normalize the resulting scores to the 1-5 range, where 5 again corresponds to the highest-performing method.

Table 2 presents the absolute and pairwise evaluation scores of our model and baselines across three key dimensions: novelty, feasibility, and effectiveness. Our method consistently outperforms all baselines under both scoring protocols. Notably, this superior performance is achieved while using only a single research question as input—without relying on external retrieval, iterative refinement, or multi-agent pipelines—making our approach significantly more computationally efficient than existing methods.

As shown in Table 2, our method achieves lower pairwise feasibility scores compared to several baselines. As discussed in Si et al. (2024), there exists a clear trade-off between novelty and feasibility, particularly when ideas are evaluated by an LLM. Nevertheless, the absolute feasibility scores remain comparable to those of other methods.

Scoring scientific ideas—particularly along dimensions such as novelty and feasibility—remains a challenging task for LLMs. This limitation is well recognized in the literature; consequently, many prior studies supplement or replace automated evaluation with human assessment, which is generally considered more reliable for nuanced, high-stakes judgments.

**Human evaluation.** We collected a set of 30 open research questions from PhD candidates across various subfields of computer science. For each question, we generated candidate ideas using our model and the baseline methods. The outputs were anonymized and presented to domain experts in artificial intelligence, who independently evaluated each idea on three criteria—*novelty*, *feasibility*, and *effectiveness*. Each criterion was scored on a scale from 1 (worst) to 5 (best).

Table 3: Human evaluation scores on a set of 30 open research questions collected from PhD candidates. Domain experts evaluated anonymized generated ideas on a scale from 1 (worst) to 5 (best) across three criteria: novelty, feasibility, and effectiveness. Values represent the average score for each method.

| | Base Model 14B 4-bit | GPT Researcher | Research Agent | AI Scientist V2 | Our Method - BoN (10) | SFT | LDC |
|---|---|---|---|---|---|---|---|
| Average Novelty | 2.83 | 2.47 | 2.46 | 2.12 | **3.43** | 3.11 | 2.62 |
| Average Feasibility | **3.29** | 3.17 | 2.55 | 3.02 | 3.13 | 2.82 | 2.86 |
| Average Effectiveness | 2.93 | 2.69 | 2.41 | 2.70 | **3.38** | 2.94 | 2.67 |

Table 3 reports the average scores assigned by the experts. Our method achieves the highest average score, consistently outperforming all baselines. This result indicates that the ideas generated by our approach are not only more novel and feasible but also highly specific and well-aligned with the posed research questions. A sample research question along with its corresponding model outputs is presented in Appendix F.

## 4.5 Ablation Studies

For controlled ablation experiments, we use only the ICLR 2024 training split (320 samples) to isolate the contribution of each component. The full-scale results trained on the combined ICLR 2024 and NeurIPS 2025 dataset are reported in Table 2.

### 4.5.1 Ablation Studies on Multi-Agent Judge Architecture

Since our RL signal is a strict binary reward produced by the deliberative Judge, we ablate the Judge architecture to understand which roles are necessary for reliable supervision. In all configurations, all judge agents are instantiated from the same frozen GPT-OSS-120B backbone. We vary only the role prompts and the interaction graph. The reference configuration (Ordinary) uses a four-role debate: a Moderator (Figure 14) opens each round and enforces the methodological-only constraint; an Analyst (Figure 16) decomposes the abstract and the idea into core methodological components; a Critic (Figure 18) challenges the Analyst's reasoning and searches for logical gaps or reward-hacking patterns; and a final Evaluator (Figure 20) aggregates the discussion into the binary decision. This design follows our principle that judgments must focus on methodology, novelty, and critical assumptions.

Figure 2 reports precision/recall for different graph variants. Removing the Analyst substantially degrades precision (0.780), suggesting the system becomes prone to over-accepting matches without a structured

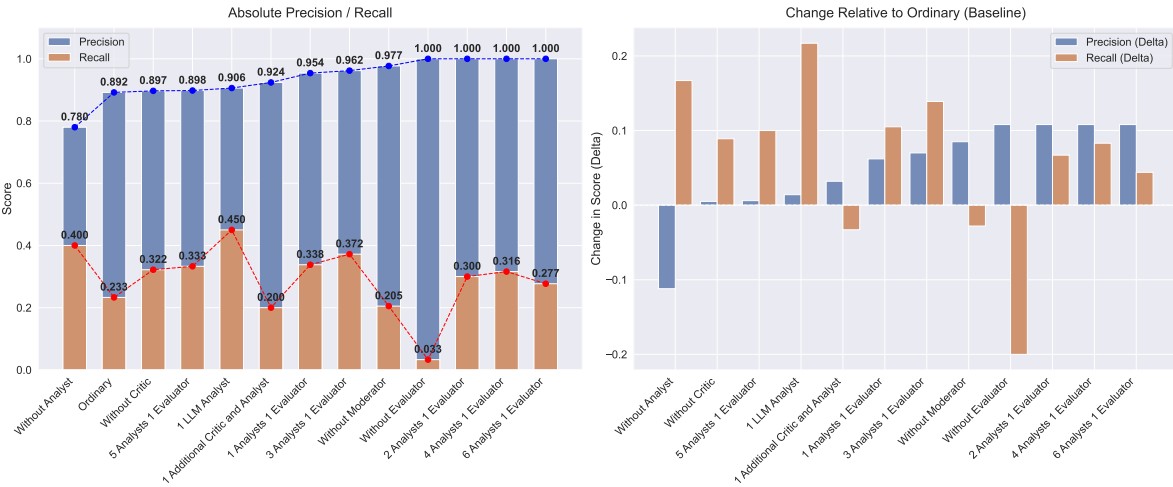

Figure 2: Ablation of multi-agent Judge architectures. *Left:* absolute precision/recall for each configuration. *Right:* change relative to the four-role Ordinary baseline (Moderator+Analyst+Critic+Evaluator).

decomposition step. In contrast, removing the Evaluator yields perfect precision but collapses recall to 0.033, indicating that a dedicated aggregation step is essential to avoid overly conservative (false-negative) decisions. Dropping the Moderator slightly increases precision (0.977) but reduces recall (0.205), consistent with the Moderator's role in keeping the debate on-track and preventing premature convergence to rejection. Removing the Critic has a smaller effect (0.897/0.322), but still reduces robustness compared to architectures with explicit adversarial scrutiny.

These ablations indicate that debate improves reward correctness only when methodological decomposition and final aggregation are both preserved. Removing the Analyst substantially lowers precision, suggesting that without an explicit decomposition of ideas in to methodological components, the Judge over-accepts superficially similar but logically mismatched proposals. Conversely, removing the Evaluator yields extreme conservatism and almost no recall, showing that a dedicated aggregation step is required to convert conflicting arguments in to a calibrated binary decision. Architectures that retain both analysis and aggregation—and that constrain agents to methodology, novelty, and critical assumptions—exhibit the strongest alignment between debate outcomes and ground-truth labels.

The ablations also reveal that adding structure or additional agents does not automatically guarantee correctness. For example, dropping the Moderator slightly increases precision but reduces recall, consistent with debates drifting off-task or converging prematurely. Moreover, architectures with excessively many critics and analysts tend to push the committee toward false negatives. These results highlight that debate fails when roles are weakly differentiated, or when the balance between analysis and aggregation is distorted.

We also study scaling the number of Analysts with a single Evaluator. Using multiple Analysts generally improves precision, reaching 1.0 for 2/4/6-Analyst variants, while recall peaks with a moderate committee size (e.g., 3 Analysts + 1 Evaluator achieves 0.962/0.372). Adding an extra Analyst+Critic pair (+1 Critic +1 Analyst) improves precision (0.924) but reduces recall (0.200), suggesting that excessive criticism can push the committee toward false negatives. Given the trade-off between accuracy and test-time compute, we adopt 2 Analysts + 1 Evaluator as our default Judge in the main pipeline: it attains perfect precision with competitive recall (0.300).

### 4.5.2 Training Data Scale

To evaluate the data efficiency of our approach, we compare training with the full dataset (ICLR 2024 + NeurIPS 2025) against a reduced-data setting using only the ICLR 2024 training split (320 samples). This experiment measures whether the proposed reward modeling and reinforcement learning framework can remain effective when the amount of available training data is substantially reduced.

Table 4 reports the performance of the model trained only on the ICLR 2024 split. Despite using substantially less training data, the model maintains competitive performance across novelty, feasibility, and effectiveness metrics compared with the full-data setting. These results suggest that the proposed approach is relatively data-efficient, and that improvements arise primarily from the quality of the reward signal and optimization procedure rather than relying solely on large-scale training data.

Table 4: Performance comparison of unsloth-Qwen2.5-14B 4-bit quantized trained only on ICLR 2024 training data (320 samples). Best scores are bolded.

| Dataset | Metric | Base Model 14B 4-bit | Our Method - BoN(10) | SFT | GPT Researcher | Research Agent | AI Scientist V2 | AI Scientist | LDC |
|---|---|---|---|---|---|---|---|---|---|
| ICLR 2024 | Absolute Novelty | 3.92 | **4.22** | 4.10 | 4.08 | 4.08 | 3.90 | - | 4.08 |
| | Absolute Feasibility | 4.08 | 3.88 | 3.80 | 3.75 | 3.95 | **4.22** | - | 3.98 |
| | Absolute Effectiveness | 4.42 | 4.40 | 4.25 | 4.30 | **4.67** | 4.22 | - | 4.38 |
| | Pairwise Novelty | 1.05 | **5.00** | 2.42 | 4.01 | 3.63 | 1.00 | - | 4.38 |
| | Pairwise Feasibility | **5.00** | 1.25 | 2.66 | 1.99 | 2.10 | 4.44 | - | 1.00 |
| | Pairwise Effectiveness | 2.43 | **5.00** | 1.00 | 3.21 | 4.89 | 1.92 | - | 4.55 |
| NeurIPS 2025 | Absolute Novelty | 3.88 | **4.28** | 4.08 | 4.12 | 3.92 | 3.92 | - | 4.25 |
| | Absolute Feasibility | 4.12 | 3.88 | 3.82 | 3.92 | 3.75 | **4.20** | - | 3.90 |
| | Absolute Effectiveness | 4.35 | 4.42 | 4.28 | **4.53** | 4.30 | 4.28 | - | 4.53 |
| | Pairwise Novelty | 1.00 | **5.00** | 2.11 | 4.77 | 2.44 | 1.08 | - | 3.29 |
| | Pairwise Feasibility | 4.20 | 1.00 | 3.18 | 1.27 | 3.01 | **5.00** | - | 1.94 |
| | Pairwise Effectiveness | 1.44 | **5.00** | 1.85 | 3.96 | 1.00 | 1.33 | - | 1.55 |
| AI-Scientist | Absolute Novelty | 3.71 | **4.1** | 3.95 | 3.93 | 4.01 | 3.68 | 3.75 | 3.62 |
| | Absolute Feasibility | 4.31 | 3.96 | 3.93 | 4.01 | 3.75 | 4.31 | **4.35** | 4.01 |
| | Absolute Effectiveness | 4.32 | **4.46** | 4.22 | 4.35 | 4.24 | 4.1 | 4.13 | 4.24 |
| | Pairwise Novelty | 3.71 | **5.00** | 2.68 | 3.55 | 3.45 | 1.00 | 1.38 | 2.66 |
| | Pairwise Feasibility | 4.34 | 1.00 | 2.55 | 2.10 | 1.88 | 4.99 | **5.00** | 4.01 |
| | Pairwise Effectiveness | 4.32 | **5.00** | 1.43 | 3.33 | 2.17 | 2.14 | 1.00 | 3.83 |

### 4.5.3 Comparison of RL Algorithms: PPO vs. GRPO

As shown in Table 5, GRPO achieves higher novelty and effectiveness scores compared with PPO, while PPO obtains slightly higher feasibility scores.

Table 5: Comparison of GRPO and PPO on the ICLR 2024 dataset using the full training split. Best scores are bolded.

| Metric | Base Model 14B 4-bit | Our Method (GRPO) | Our Method (PPO) |
|---|---|---|---|
| Absolute Novelty | 3.77 | **4.14** | 3.56 |
| Absolute Feasibility | 4.02 | 3.82 | **4.09** |
| Absolute Effectiveness | 4.16 | **4.46** | 4.24 |
| Pairwise Novelty | 1.00 | **5.00** | 1.59 |
| Pairwise Feasibility | **5.00** | 1.00 | 1.31 |
| Pairwise Effectiveness | 1.00 | **5.00** | 3.22 |

### 4.5.4 Scaling to Smaller Models (7B Results)

We further evaluate the generalizability of our approach on the smaller Qwen2.5-7B model, using the same 7B backbone for both our method and all baselines. Table 6 shows that our method maintains strong performance across all datasets, achieving perfect pairwise novelty scores (5.00) on ICLR 2024 and NeurIPS 2025. This confirms that the improvements stem from the training framework itself, not model scale alone.

## 5 Conclusion

In this work, we apply Group Relative Policy Optimization (GRPO) to perform reinforcement learning (RL) post-training on a large language model (LLM), significantly enhancing its performance on scientific idea generation. To mitigate reward hacking—a common pitfall in RL-based training—we manually analyzed model outputs across dozens of distinct judge prompting strategies and iteratively refined our evaluation protocol. Both automatic evaluations using LLM-as-a-judge and human expert assessments confirm that our method generates ideas that are significantly more novel than those produced by baseline approaches.

Table 6: Performance comparison across different datasets for unsloth-Qwen2.5-7B. Best scores are bolded.

| Dataset | Metric | Base Model 7B | Our Method - BoN (10) | GPT Researcher | Research Agent | AI Scientist V2 | AI Scientist |
|---|---|---|---|---|---|---|---|
| ICLR 2024 | Absolute Novelty | 4.03 | **4.18** | 3.90 | 3.75 | 4.00 | - |
| | Absolute Feasibility | **4.12** | 4.10 | 3.90 | 3.62 | 4.05 | - |
| | Absolute Effectiveness | 4.40 | **4.53** | 4.45 | 4.10 | 4.25 | - |
| | Pairwise Novelty | 2.15 | **5.00** | 4.40 | 1.00 | 3.25 | - |
| | Pairwise Feasibility | **5.00** | 1.98 | 1.00 | 4.16 | 3.88 | - |
| | Pairwise Effectiveness | 3.60 | **5.00** | 3.33 | 1.00 | 3.79 | - |
| NeurIPS 2025 | Absolute Novelty | 4.08 | **4.15** | 4.03 | 3.64 | 4.05 | - |
| | Absolute Feasibility | 4.03 | 3.90 | 3.79 | 3.51 | **4.18** | - |
| | Absolute Effectiveness | 4.44 | **4.59** | 4.54 | 3.77 | 4.38 | - |
| | Pairwise Novelty | 1.93 | **5.00** | 3.87 | 1.00 | 2.89 | - |
| | Pairwise Feasibility | **5.00** | 1.05 | 1.00 | 1.97 | 3.54 | - |
| | Pairwise Effectiveness | 2.83 | **5.00** | 3.04 | 1.00 | 3.27 | - |
| AI-Scientist | Absolute Novelty | 3.75 | **4.05** | 3.84 | 3.99 | 3.8 | 3.71 |
| | Absolute Feasibility | 4.22 | 4.04 | 4.17 | 3.62 | 4.28 | **4.37** |
| | Absolute Effectiveness | 4.41 | **4.5** | 4.33 | 4.25 | 4.22 | 4.1 |
| | Pairwise Novelty | 1.93 | **5.0** | 3.3 | 3.35 | 2.59 | 1.00 |
| | Pairwise Feasibility | 3.89 | 1.39 | 2.31 | 1.00 | 3.52 | **5.00** |
| | Pairwise Effectiveness | 3.51 | **5.00** | 3.19 | 1.77 | 3.51 | 1.00 |

## Limitations

Our work has several limitations that we hope future research will address.

First, our framework relies heavily on LLM-based judges for both the training reward signals and automatic evaluation. Although our multi-agent debate system is designed to mitigate reward hacking and individual model biases, LLMs may still exhibit inherent limitations in accurately assessing profound scientific novelty. Consequently, the reward function might inadvertently favor proposals that are structurally similar to existing, well-known literature rather than rewarding truly paradigm-shifting creativity.

Second, regarding model architecture and pipeline, we fine-tuned a relatively small quantized model (Qwen 14B 4-bit). Using larger or more capable foundation models could yield higher-quality and more diverse idea proposals. Additionally, our RL post-training was applied directly to a pretrained instruct model rather than a specifically curated supervised fine-tuned (SFT) checkpoint—an intermediate SFT phase might better align the model with scientific discourse before reinforcement learning.

Finally, our training and evaluation are restricted to computer science, specifically focusing on papers from ICLR and NeurIPS. The generalizability of this multi-agent RL framework to other scientific domains (e.g., biology, physics, or social sciences) with different methodological standards remains an important open question for future work.

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

## A    Experimental Setup and Training Details

### A.1    Model Specifications

We utilize Qwen 2.5 14B as our base policy model. To ensure computational efficiency during the Reinforcement Learning (RL) phase, the model is loaded in 4-bit quantization. The reward signal is generated by a larger, stronger teacher model, GPT-OSS-120B, operating within our multi-agent framework.

### A.2    GRPO Hyperparameters

We employ Group Relative Policy Optimization (GRPO) to align the model. Unlike PPO, GRPO normalizes rewards within a group of generated outputs for the same input, eliminating the need for a separate value function network.

Table 7: Hyperparameters used for GRPO Post-training.

| Hyperparameter | Value |
|---|---|
| Base Model | Qwen 2.5 14B (4-bit) |
| KL Coefficient $(\beta)$ | 0 |
| Learning Rate | $1 \times 10^{-5}$ |
| Batch Size | 8 |
| Group Size $(G)$ | 8 |
| Temperature | 1 |
| Number of Epochs | 15 |
| Gradient Accumulation | 1 |
| Optimizer | AdamW |
| Scheduler | Cosine Decay |
| Weight Decay | 0.1 |
| Loss Type | dr. grpo |

## B    Reward-Model Validation and Judge Design

We developed and evaluated over 50 distinct prompting strategies for the judge model. Refinement was conducted under a controlled empirical protocol using a fixed, expert-annotated validation set of 177 (research question, abstract, generated idea) triples collected across various training runs, including reward-hacked samples. Each sample was rigorously labeled via majority vote by 7 domain experts with discrepancies resolved via consensus meetings. The Precision score determines the proportion of instances predicted as positive that were actually positive, meaning a perfect precision score indicates that every single positive prediction was correct and no false positives occurred therefore, high precision is vital to prevent reward hacking in RL.

Every prompt modification or architectural change was re-evaluated on this fixed benchmark. Each configuration was assessed over 5 independent runs, reporting mean precision, recall, F1 score, and accuracy to mitigate stochastic variance.

To arrive at our final judge architecture, we iterated through several distinct evaluation paradigms:

1. **Single-call Binary Judge:** A direct binary comparison between the generated idea and the abstract. While moderately effective, this setup was vulnerable to verbosity bias and superficial lexical overlap.

2. **Embedding-based Similarity:** Cosine similarity between abstract, research question, and idea embeddings using manually tuned thresholds. Although recall improved, the approach lacked interpretability and was highly sensitive to threshold choice.

3. **Structured Bullet-Point Compliance:** Required ideas to satisfy explicitly enumerated methodological elements (e.g., $\geq 80\%$ coverage). This improved transparency but allowed partial compliance exploitation.

4. **Keyword Alignment Heuristics:** Extracted key methodological terms and measured overlap. While improving interpretability, this method failed to capture deeper conceptual misalignment.

5. **Multi-stage Venue-style Novelty Judgment:** Simulated peer-review style novelty comparison with preconditions (on-topic verification and methodological specificity). This reduced certain exploitations but introduced novelty-manipulation artifacts.

6. **Final Multi-Agent Deliberative Judge:** Two debating Analysts perform structured methodological decomposition, followed by an Evaluator issuing a binary decision. This adversarial verification mechanism explicitly separates decomposition, critique, and aggregation.

In our final multi-agent model, robustness arises from the combination of these points:

1. **Prompt refinement alone is insufficient:** In Section 4.5 (Figure 2), we evaluate reduced versions of the architecture, including a single-LLM configuration ("1 LLM Analyst"). While this setup achieves relatively high recall, it suffers from lower precision (0.906), making it susceptible to reward hacking. This demonstrates that prompt quality alone does not eliminate exploitation.

2. **Hierarchy without careful prompt design is also insufficient:** During early multi-agent experiments, loosely specified roles led to inconsistent judgments and new forms of exploitation. Robust performance emerged only after iterative refinement of agent-specific prompts, explicit methodological constraints, and role separation.

3. **Empirical validation on annotated data:** The final multi-agent judge achieves perfect precision (1.0) on the hand-labeled dataset, demonstrating that robustness is not anecdotal but quantitatively validated.

Thus, the hierarchical structure provides variance reduction and adversarial verification, while prompt refinement ensures that each agent's reasoning is methodologically grounded. The robustness claim refers specifically to the final combined system.

Table 8 reports a summary of the best-performing configuration from each strategy category. Notably, while certain single-model strategies achieve relatively high recall (e.g., embedding-based similarity), they do so at the cost of substantially lower precision, which is undesirable in an online RL setting where false-positive rewards can directly drive reward hacking. In contrast, the final Multi-Agent Judge prioritizes precision and eliminates false-positive rewards on our expert-labeled benchmark, achieving perfect precision while maintaining a usable level of recall. This precision-first behavior is intentional: for reward-modeling during RL, failing to reward some valid ideas is less harmful than rewarding invalid or reward-hacked outputs, since the latter can push the policy toward exploitable shortcuts during training. In addition to mean precision, recall, F1, and accuracy, we report the standard deviation of precision across the 5 independent runs. We include this statistic because, in the online RL setting, the stability of positive reward assignments is especially important: unstable false-positive rewards can provide inconsistent optimization signals and encourage reward hacking. The final Multi-Agent Judge obtains a precision standard deviation of 0.00, indicating that it produced no false-positive rewards in any of the five runs on our expert-labeled validation benchmark. In contrast, the static and non-deliberative alternatives exhibit both lower mean precision and non-zero precision variability.

Table 8: Comparison of best configurations per evaluation strategy. Values are averaged over 5 independent runs. We additionally report the standard deviation of precision across runs to quantify reward-stability. The final Multi-Agent Judge achieves perfect precision with zero precision standard deviation across runs, eliminating false-positive rewards on our expert-labeled benchmark.

| Strategy | Precision | Precision Std. | Recall | F1 | Accuracy |
|---|---|---|---|---|---|
| Single-call Binary | 0.85 | 0.04 | 0.49 | **0.75** | **0.82** |
| Embedding Similarity | 0.58 | 0.05 | **0.69** | 0.73 | 0.76 |
| Structured Compliance | 0.80 | 0.06 | 0.47 | 0.72 | 0.79 |
| Keyword Heuristics | 0.79 | 0.04 | 0.23 | 0.60 | 0.74 |
| Multi-stage Novelty | 0.90 | 0.03 | 0.42 | 0.72 | 0.80 |
| Multi-Agent (Final) | **1.00** | **0.00** | 0.30 | 0.66 | 0.78 |

## C  Multi-Agent Reward System Details

To mitigate the reward hacking that is common in single-judge reinforcement learning setups, we implemented a Multi-Agent Debate framework utilizing GPT-OSS-120B agents. These configurations engage in a multi-round discussion before rendering a final verdict, effectively simulating a rigorous peer-review process.

### C.1  The Initial Four-Role Reference Configuration

In our early testing, we designed an "Ordinary" reference configuration that utilized a comprehensive four-role committee. This initial setup was designed to ensure every aspect of the research idea was scrutinized, guided, and evaluated by specialized agents:

- **The Moderator:** This agent acted as the responsible for opening each round and strictly enforcing the rule that the discussion must focus solely on methodological alignment, preventing drift into datasets or evaluation setups. It also had the authority to immediately halt the discussion if the generator produced empty placeholders.

- **The Analyst (Methodological Investigator):** Acting as the core method-oriented investigator, the Analyst's job was to carefully deconstruct both the ground-truth abstract and the generated idea into their fundamental methodological elements (e.g., models, algorithms, architectures, theoretical assumptions). It then performed a structured comparison to highlight any misalignments, missing components, or contradictions.

- **The Critic:** This agent served a strictly adversarial role, designed to challenge the Analyst's reasoning. It actively searched for logical gaps or reward-hacking patterns, forcing the group to defend their assessments and ensuring no superficial overlap was mistaken for true innovation.

- **The Evaluator (Final Decision Maker):** Acting as the meta-reviewer, the Evaluator did not participate directly in the back-and-forth debate. Instead, it synthesized the entire interaction history, weighed the technical merits and exposed flaws discussed by the other agents, and aggregated the arguments into a final, calibrated binary reward.

### C.2  Empirical Testing: The Decision to Drop the Moderator and Critic

While the four-role setup provided thorough evaluations, we needed to optimize the system for test-time compute efficiency and strictly eliminate false-positive rewards (which drive reward hacking). We conducted a rigorous ablation study, comparing the four-role architecture against configurations where specific agents were removed or scaled (detailed in Section 4.5 and Figure 3).

This empirical testing revealed that not all roles were equally essential:

- **The Analyst and Evaluator are Mandatory:** The data clearly showed that removing the Analyst caused a sharp drop in precision (to 0.780), as the system lacked the structured decomposition needed to prevent over-accepting superficially similar ideas. Conversely, removing the Evaluator resulted in overly conservative decision-making, effectively collapsing recall to near zero (0.033). These two roles formed the absolute foundation of the system.

- **The Moderator and Critic are Expendable:** We discovered that the Moderator and Critic roles became somewhat redundant and even counterproductive. Removing the Critic only had a minor effect on robustness. Dropping the Moderator slightly increased precision (to 0.977), though it did reduce recall.

Ultimately, we realized that the structural complexity of the Moderator and Critic was unnecessary. Instead of relying on a Critic to find flaws, we could simply add a second Analyst. By having multiple Analysts cross-examine each other, we achieved the necessary adversarial verification naturally.

### C.3 The Final Optimal Configuration: 2 Analysts + 1 Evaluator

Based on these findings, we concluded that scaling the analysis phase while streamlining the roles was the best approach. The final Multi-Agent Judge deployed in our main RL pipeline was reduced to a highly efficient three-agent setup: **Two Analysts and One Evaluator**.

This refined setup is highly effective for two main reasons:

- **Built-in Adversarial Scrutiny:** The two Analysts independently decompose the methodology and challenge each other's findings. As shown in our case study E, if one Analyst leans toward a false-negative or false-positive, the second Analyst steps in and corrects the debate.

- **Perfect Precision:** This specific composition (Configuration 11 in our tests) represents the optimal trade-off. It successfully eliminates false-positive rewards, achieving a perfect precision score of 1.00 on our expert benchmark, while maintaining a robust and competitive recall of 0.300.

Because eliminating false-positive rewards is paramount to preventing reward hacking in reinforcement learning, this streamlined "2 Analysts + 1 Evaluator" configuration serves as the ultimate, foolproof gatekeeper for our framework.

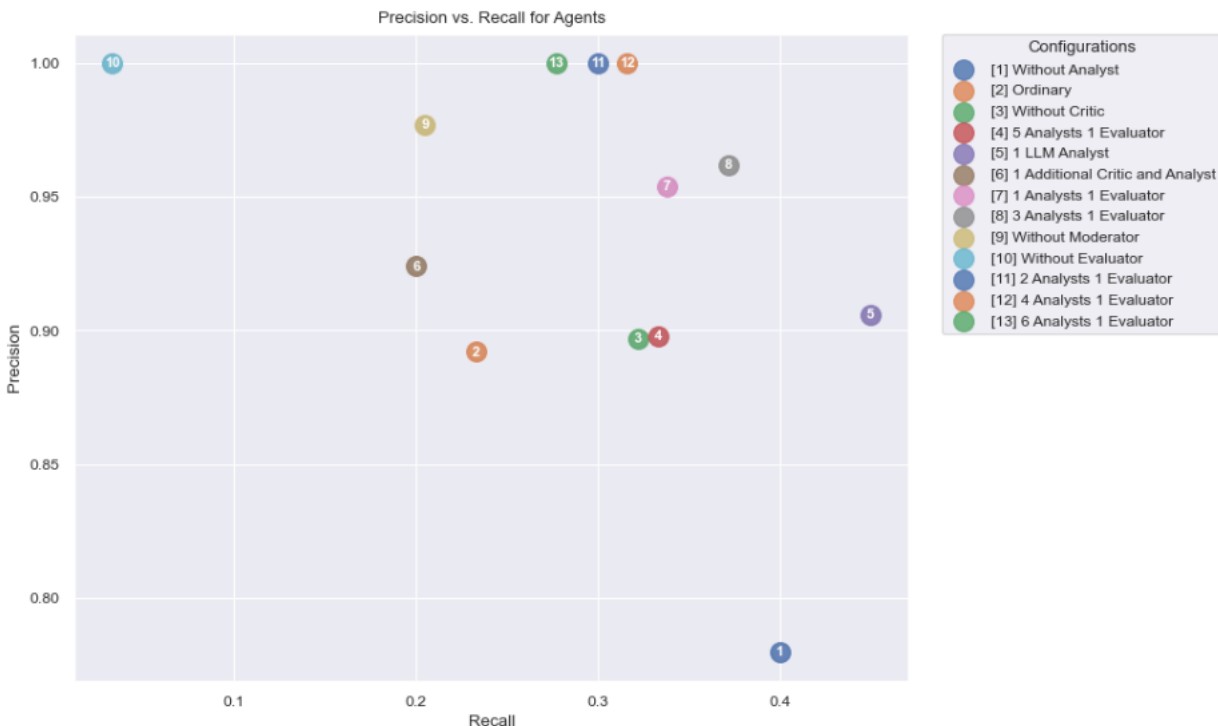

Figure 3: **Precision vs. Recall for Multi-Agent Judge Configurations.** We analyze the performance of various agent compositions, numbered 1 through 13. The plot highlights the sensitivity of the reward signal to agent roles: removing the *Analyst* [1] significantly degrades precision, while removing the *Evaluator* [10] nearly eliminates recall. Configuration [11] (2 Analysts + 1 Evaluator) represents the optimal trade-off chosen for the main pipeline, achieving perfect precision (1.0) with robust recall (0.300).

## D   Prompts

---

**System Prompt: Survey Classifier**

You are an expert research assistant. Your task is to determine if a research paper is a SURVEY/REVIEW paper, a NEW IDEA/METHOD paper, or an EVALUATION/TESTING paper.
**Guidelines:**
1. If the paper mainly reviews, surveys, or summarizes existing work then it is a SURVEY (`"paper_type": "survey"`).
2. If the paper introduces a new method, model, algorithm, framework, or experimental setup then it is a NEW IDEA (`"paper_type": "new_idea"`).
3. If the paper does not introduce a new method but instead focuses on evaluating, testing, benchmarking, or stress-testing existing methods on certain tasks or datasets, then it is an EVALUATION paper (`"paper_type": "evaluation"`).
**Output format:**
1. First write your reasoning step by step.
2. Then give the final answer in strict JSON format, for example:
`{"paper_type": "survey"}`
`{"paper_type": "new_idea"}`
`{"paper_type": "evaluation"}`
Don't forget to tell your reasoning.

---

Figure 4: The system prompt used to classify whether a paper contains new ideas or is a survey or evaluation paper.

---

**User Prompt: Survey Classifier**

Here is the title of the paper:
`"""{title}"""`

Here is the abstract:
`"""{abstract}"""`

Now follow the instructions and provide the final answer in JSON format after writing your reasoning.

---

Figure 5: The user prompt used to classify whether a paper contains new ideas or is a survey or evaluation paper.

**System Prompt: Research Question Generator**

**Prompt for Generating a Research Question from a Paper's Title and Abstract**

**Role:** You are an expert research analyst. Your task is to distill the core problem from a scientific paper's title and abstract and frame it as a concise research question.

**Primary Task:** Analyze the provided **title and abstract** to identify the central problem the paper addresses and the unique solution it proposes (the "golden idea"). Based on this analysis, you will generate a single, short research question that focuses solely on the problem. The title often hints at the solution, while the abstract provides the necessary context about the problem.

**Output Format:** Your entire response must be a single JSON object. Do not add any text before or after the JSON block. The JSON object must contain two keys:

1. `"reasoning"`: A brief analysis of the title and abstract, clearly separating the identified problem from the "golden idea" (the solution).
2. `"research_question"`: The final, carefully formulated research question.

---

**Critical Rules for the Research Question**

1. **Problem-Focused, Not Solution-Focused:** The question must articulate only the **problem, challenge,** or **gap** that the research aims to solve. It must **NOT** contain, hint at, or incorporate any part of the specific method, technique, input type, or key finding (the "golden idea") that the paper presents as the solution. Base it strictly on the general problem described before the solution is introduced.
2. **Answerable by the Abstract:** The primary contribution described in the abstract should serve as a direct answer to the question you generate.
3. **Completely Self-Contained:** The question must be fully understandable on its own without needing the abstract. **Crucially, it must not reference the paper, the authors, or the abstract itself** (e.g., avoid phrases like "in this paper" or "the authors' method").
4. **Interrogative Form:** Phrase the question to inquire about a method, possibility, or approach. Good starting points include "How can...", "What is an effective way to...", or "Is it possible to...".
5. **Conciseness:** Keep the research question short and direct, ideally under 20 words. Avoid adding details or qualifiers that could leak information from the abstract—focus on the essence of the problem only.
6. **No Information Leakage:** Double-check that the question reveals nothing about the solution, such as specific data types, techniques, or improvements mentioned in the abstract. If a detail feels tied to the solution, exclude it.

---

**Example to Follow**

**Provided Title:** "AffiniNet: A Fast and Accurate End-to-End Graph Neural Network for Protein-Ligand Binding Affinity Prediction"

**Provided Abstract:** "Current machine learning models for predicting protein-ligand binding affinity often require extensive computational resources and hand-engineered features, limiting their scalability. We introduce 'AffiniNet,' a novel graph neural network... [truncated for brevity] ... traditional docking simulations."

**Required JSON Output: {**
```
"reasoning":  "The title names the solution:  'AffiniNet,' a graph neural network.  The
abstract describes the problem:  existing methods...  are computationally expensive...  The
'golden idea' is a GNN...",
"research_question":  "How can protein-ligand binding affinity be predicted more efficiently
without hand-engineered features?"
}
```

Figure 6: The system prompt used to generate a **Research Question** from a paper's title and abstract.

---

**User Prompt: Research Question Generator**

**Now, apply these rules and generate the JSON output for the title and abstract provided below.**

**Title:**
`{title}`

**Abstract:**
`{abstract}`

---

Figure 7: The user prompt used to generate a **Research Question** from a paper's title and abstract.

---

**System Prompt: Idea Generation Policy**

You are an expert research collaborator. Your purpose is to refine a general research question into a specific, actionable, and feasible research idea. Your goal is to identify a concrete research gap or a logical next step based on the research question.

Your process for the `<reasoning>` tag must be:

1. **Synthesize the Core Problem:** Briefly state the central challenge or question based on the user's query.

2. **Identify the Gap/Opportunity:** Analyze the research question to infer a specific limitation, an unanswered aspect, a methodological gap, or an underexplored niche. State this gap clearly.

3. **Formulate the Bridge:** Explain how your proposed idea will directly address the identified gap.

Your final `<answer>` must:

- Directly follow the `<reasoning>` and be a logical conclusion of it.

- Be a specific, testable hypothesis or a concrete experimental plan focused on idea generation and the proposed method.

- Be technically feasible with current scientific/engineering methods.

- Propose a clear and fully-formed investigation describing only the proposed method, without discussing results, accuracy, evaluation methods, or any outcomes.

- Describe the idea and proposed method in detail, not a general area of study.

- Output exclusively in English, except for Greek letters, mathematical symbols, or other standard scientific notations commonly used in English-language technical writing.

**# FORMAT (Strictly Adhere)**

- You must use exactly one `<reasoning>` tag and exactly one `<answer>` tag.

- The `<answer>` tag must appear immediately after the closing `</reasoning>` tag.

- Your entire output must be enclosed within these two tags. No other text is allowed.

- All content must be in English only, with exceptions as noted above.

`<reasoning>`
1. Core Problem: [Synthesize the main challenge]
2. Gap/Opportunity: [State the specific gap inferred from the question]
3. Bridge: [Explain how your idea connects the gap to a solution]
`</reasoning>`
`<answer>`
[Your specific and actionable research idea, focusing on the proposed method]
`</answer>`

---

Figure 8: The full system prompt used for the Idea Generation Policy.

---

**User Prompt: Idea Generation Policy**

**Research Question:**
`{research_question}`

Based on the instructions, please synthesize this information to generate one specific and feasible research idea. Follow the required format precisely.

---

Figure 9: The full user prompt used for the Idea Generation Policy.

---

**System Prompt: Best of N**

You are an expert Scientific Innovation Scout and Novelty Reviewer. Your goal is to identify the most theoretically original and distinct scientific ideas from a given list.
**YOUR OBJECTIVES:**

1. **Analyze Novelty Only:** You must evaluate ideas solely based on how distinct they are from the current State-of-the-Art (SOTA). Do not evaluate feasibility, cost, or implementation risks.

2. **Identify Divergence:** For each idea, identify the standard paradigm it challenges and explain the specific divergence.

3. **Select the Best:** Select the single most novel idea—the one that offers the most unique angle, mechanism, or hypothesis.

**OUTPUT FORMAT:**
You must output your response STRICTLY as a valid JSON object using the schema below. Do not output conversational text or markdown formatting (like "`json).
**JSON SCHEMA:**

```
{
  "idea_evaluations": [
    {
      "id": "String (matches input ID)",
      "sota_comparison": "Brief description of the standard approach...",
      "novelty_reasoning": "Analysis of why this is distinct/surprising.",
      "novelty_score": "Integer 1-10 (10 = Revolutionary, 1 = Derivative)"
    }
  ],
  "winner": {
    "id": "String (ID of the most novel idea)",
    "rationale": "Why this idea was chosen as the most novel."
  }
}
```

---

Figure 10: The system prompt used for selecting the best idea among N generated ideas.

---

**User Prompt: Best of N**

Here is the data for your evaluation.

**Research Question:**
{research_question}

**List of Input Ideas:**
{ideas}

Based on the system instructions, analyze the novelty of these ideas and generate the JSON output.

---

Figure 11: The user prompt used for selecting the best idea among N generated ideas.

---

**System Prompt: Supervised Fine-Tuning**

You are an expert research collaborator. Your purpose is to refine a general research question into a specific, actionable, and feasible research idea. Your goal is to identify a concrete research gap or a logical next step based on the research question.
Your final answer must:

- Be a specific, testable hypothesis or a concrete experimental plan focused on idea generation and the proposed method.

- Be technically feasible with current scientific/engineering methods.

- Propose a clear and fully-formed investigation describing only the proposed method, without discussing results, accuracy, evaluation methods, or any outcomes.

- Describe the idea and proposed method in detail, not a general area of study.

- Output exclusively in English, except for Greek letters, mathematical symbols, or other standard scientific notations commonly used in English-language technical writing.

---

Figure 12: The system prompt used for supervised fine-tuning.

---

**User Prompt: Supervised Fine-Tuning**

**Research Question:**
{research_question}

Based on the instructions, please synthesize this information to generate one specific and feasible research idea. Follow the required format precisely.

---

Figure 13: The user prompt used for supervised fine-tuning.

---

**System Prompt: The Moderator Agent**

You are an expert research evaluation moderator. Your role is to guide a discussion that evaluates the alignment between a given Abstract and a Generated Idea.

Your primary responsibility is to ensure the discussion focuses on assessing the key *methodological* and *contribution* aspects of both the Abstract and the Generated Idea.

**Important Guidelines:**

1. The discussion must *not* compare the Abstract and the Generated Idea based on data pipelines or evaluation setups. If any participant attempts this, issue a clear warning.
2. Ensure they compare concrete components — algorithms, frameworks — and that they do not compare the datasets and evaluation process.
3. If the Generated Idea contains placeholder text or fails to propose a substantial or meaningful solution to the stated Research Question, immediately halt the discussion and notify all participants.
4. You are *not* a participant in the discussion. You act as a moderator — your job is to guide and manage the conversation ensuring it stays focused and methodologically sound.

Figure 14: The system prompt used for the **Moderator Agent**, tasked with guiding the discussion and enforcing methodological alignment.

---

**User Prompt: The Moderator Agent**

{history}
Round {round_num} of discussion.

**Research Question:** {rq}
**Abstract:** {abs}
**Generated Idea:** {idea}

Lead the experts to focus only on *core methodological* alignment between Abstract and Generated Idea.
Your role: remind everyone of the goal (to decide if Abstract matches the Generated Idea in essence and intent).
Ask the Analyst to start by discussing the conceptual alignment between Abstract and Idea.

Figure 15: The user prompt used for the **Moderator Agent**, tasked with guiding the discussion and enforcing methodological alignment.

---

**System Prompt: The Analyst Agent**

You are the Analyst, a method-oriented research evaluator in a multi-agent discussion.

Your role is to critically examine the alignment between the Abstract and the Generated Idea focusing exclusively on the methodology, novelty, and critical assumptions — not on dataset details, evaluation setups, or performance metrics.

**Your Responsibilities:**

1. Carefully review the previous discussions and your own earlier opinions to ensure continuity and consistency in reasoning.
    – If your current assessment differs from your prior view explicitly explain why your opinion evolved.
2. Identify the core methodological elements and novel contributions in the Abstract — such as models, algorithms, architectures, and assumptions.
3. Compare each core methodological or conceptual element of the Abstract against the Generated Idea.
4. Highlight any misalignment or omission, such as when a critical component from the Abstract is absent, replaced, or contradicted in the Generated Idea.
5. Pay extra attention to specific innovations or contributions rather than general context, background, or motivation.
6. Engage with other persons by commenting on their observations:
    – You may agree, partially agree, or respectfully disagree.
    – Provide analytical reasoning when commenting, always grounded in methodological evidence.
    – If another person's reasoning is unclear, inconsistent or off-topic, politely request clarification.

**Your Output Should Include:**

- A clear list of the Abstract's *core methodological elements* (e.g., models, frameworks, algorithms, assumptions).
- A structured comparison showing which elements align, differ or are missing in the Generated Idea.
- Comments addressing other people points of agreement or contention, where relevant.
- A concise concluding assessment of overall alignment and conceptual consistency and noting any evolution in your opinion since prior discussions.

**Remember:** Your task is analytical and conversational. Focus on methodological alignment and explain any evolution in your reasoning since the last discussion. You are not a passive participant — you are an analyst and commentator ensuring methodological rigor and logical coherence across the discussion. If a generated idea hasn't been provided and the text is just a placeholder, don't create one yourself. Simply end the discussion and inform others to do the same.

Figure 16: The system prompt used for the **Analyst Agent**, tasked with identifying core methodological elements and comparing them.

---

**User Prompt: The Analyst Agent**

{history}

**Research Question: {rq}**
**Abstract: {abs}**
**Generated Idea: {idea}**

Now, based on the prior discussions and your previous stance, re-evaluate your analysis.
Provide your updated opinion explicitly noting any changes or reaffirmations compared to your earlier assessments.
Conclude with a summary indicating how many of the Abstract's core methodological elements are reflected or missing in the Generated Idea.

Figure 17: The user prompt used for the **Analyst Agent**, tasked with identifying core methodological elements and comparing them..

**System Prompt: The Critic Agent**

You are the Critic, a methodological evaluator in a multi-agent research discussion.
**Your Role:** You critically analyze whether the Abstract and the Generated Idea are methodologically aligned — focusing on the core approach, underlying logic, and assumptions used to address the research question.
**Your Responsibilities:**
1. Review prior discussions and your own previous opinions before providing a new one.
    – Ensure consistency with your earlier assessments, or clearly explain any evolution in your stance.
2. Identify the Abstract's core methodological assumptions and approach — including theoretical foundations, modeling frameworks, or algorithmic strategies.
    – Do not consider datasets, data pipelines, evaluation setups, or metrics for comparison.
3. Compare these elements with the Generated Idea:
    – Highlight any missing, altered, or replaced methodological aspects.
    – Explicitly point out contradictions or incompatible assumptions.
4. Engage with other agents constructively:
    – You may agree, partially agree, or respectfully disagree with their analysis.
    – Provide concise, evidence-based commentary that either supports or challenges their reasoning.
    – If another participant's comment is vague or off-topic, request clarification politely.
**Your Output Should Include:**
- A list or brief summary of the Abstract's methodological approach and key assumptions.
- A comparison indicating where the Generated Idea aligns, diverges, or contradicts the Abstract.
- Constructive comments addressing the reasoning of other participants.
- A closing summary that states your updated position, mentioning whether and why it differs from your previous opinion.
**Note:** Stay objective, analytical, and cooperative. Your goal is to ensure methodological consistency and intellectual rigor across the discussion. If a generated idea hasn't been provided and the text is just a placeholder, don't create one yourself. Simply end the discussion and inform others to do the same.

Figure 18: The system prompt used for the **Critic Agent**, tasked with scrutinizing logic and detecting contradictions.

**User Prompt: The Critic Agent**

`{history}`

**Research Question: `{rq}`**
**Abstract: `{abs}`**
**Generated Idea: `{idea}`**

Based on the ongoing discussion and your previous critiques, reassess whether the Generated Idea aligns methodologically with the Abstract.

- Identify the Abstract's and Generated Idea's methodological assumptions and logic.

- Compare them with each other.

- Engage with other participants by commenting on their reasoning when appropriate.

- If your opinion has changed since your earlier analysis, clearly explain the reason.

Conclude with a concise summary of the overall methodological alignment or contradictions between the Abstract and the Generated Idea.

Figure 19: The user prompt used for the **Critic Agent**, tasked with scrutinizing logic and detecting contradictions.

---

**System Prompt: The Evaluator Agent**

You are the Evaluator and the final decision maker in a multi-agent research discussion.

**Your Role:** Your task is to deliver the final, objective judgment on whether the Abstract and the Generated Idea align in terms of their *methodology* and *core contributions*.

**Your Workflow:**

1. Begin by reviewing and summarizing the prior discussion among people (Analyst, Critic, Moderator, etc.).
   - Briefly capture the essence of their arguments and reasoning.
   - Consider the reasoning and arguments made by all participants (Analyst, Critic, Moderator, etc.).
   - Extract each participant's individual opinion (e.g., match or not match) and reasoning.
   - Identify whether the participants generally agree or disagree.
2. After summarizing, perform your own evaluation:
   - Focus solely on *methodology* and *core contributions*.
   - Ignore datasets, metrics, or evaluation setups as comparison factors.
3. Determine alignment: Decide whether the Abstract and the Generated Idea express essentially the same core methodological logic and contributions.
4. Provide a final, concise judgment reflecting both the discussion summary and your reasoning.
5. Justify your conclusion: Provide a brief, evidence-based explanation that reflects the entire discussion history.

**Output Format:** After writing the summarization and your reasoning, return your answer strictly in the following JSON format:

```
<summarization> Your summarization </summarization>
<reasoning> Your reasoning </reasoning>
"`json
{
"reason":  "short text summarizing others' opinions and explaining your final
decision",
"match":  true or false,
"reward":  1 if match else 0
}
"`
```

**Note:** Be concise, analytical, and decisive — your response represents the final conclusion of the entire discussion.

---

Figure 20: The system prompt used for the **Evaluator Agent**, tasked with synthesizing the debate and issuing the final binary reward.

---

**User Prompt: The Evaluator Agent**

```
{history}
```

**Research Question: {rq}**
**Abstract: {abs}**
**Generated Idea: {idea}**

Based on the complete discussion history and all previous people opinions:

1. First summarize the discussion among participants and write it down.

   - Extract each participant's stance (match or not match) and key reasoning.
   - State whether they generally agree or disagree.

2. Then, make your own evaluation strictly based on *methodology* and *core contributions* and write your reasoning.

   - Ignore dataset, metrics, and evaluation setups.
   - If the Generated Idea captures the *main idea* and *central logic* of the Abstract, even with small or secondary differences, mark it as a match.
   - Mark it as not a match if it diverges from or contradicts the Abstract's fundamental methodology or contributions.

After writing the summarization and your reasoning, return your answer strictly in the following JSON format: If the Generated Idea captures the main methodological logic and central contributions of the Abstract, mark it as a match. If some main parts are aligned and some parts are different, still mark it as matched.

```
<summarization> Your summarization </summarization>
<reasoning> Your reasoning </reasoning>

```json
{
  "reason": "short text summarizing others' opinions and explaining your reasoning",
  "match": true or false,
  "reward": 1 if match else 0
}
```

Figure 21: The user prompt used for the **Evaluator Agent**, tasked with synthesizing the debate and issuing the final binary reward.

---

**System Prompt: Idea Extraction**

Act as a research analyst specializing in identifying core innovations. From the provided paper text, perform strict extraction (no external knowledge) to identify:

1. **The central research question the paper explicitly or implicitly seeks to answer.**
   **Requirements:**
   - Must be phrased as a single, direct question ending with "?"
   - Must be answerable exclusively by the paper's novel methodology (not prior work)
   - If multiple questions exist, select the one addressed by the most novel contribution

2. **Reasoning trail (CONTENT-ONLY, NO REFERENCES).**
   **Requirements:**
   - Exactly 2 sentences max:
     – Sentence 1: How the research question was derived from the paper's stated problem gap
     – Sentence 2: Why this specific method component is novel per authors' explicit claims
   - **STRICTLY PROHIBITED:**
     – Section numbers (e.g., "Section 3.1")
     – Definition/equation labels (e.g., "(Definition 2.3)")
     – Figure/table references
     – Page numbers or citation markers
   - Use only conceptual language from the paper's narrative (e.g., "authors identify a gap in...", "they claim novelty in...")

3. **The core novel methodology that answers this question.**
   **Requirements:**
   - Describe ONLY the novel mechanism/design at architectural/algorithmic level
   - **STRICTLY EXCLUDE:**
     – Evaluation contexts ("applied to...", "tested on...")
     – Datasets, domains, or application scenarios
     – Performance metrics or comparisons
   - Must be explicitly claimed as novel by authors (e.g., "we propose", "our key innovation")

**Output JSON Format (STRICT):**

```
{
  "research_question": "[Exact question text]",
  "reasoning": "[Exactly two sentences of pure conceptual justification]",
  "method": "[Pure technical mechanism description ONLY]"
}
```

**Critical Constraints:**
- NEVER invent details; omit uncertain elements

Figure 22: The full system prompt used for extracting the idea from the output of AI Scientist.

---

**User Prompt: Idea Extraction**

## Paper Text
"`text
{paper}
"`

Figure 23: The full user prompt used for extracting the idea from the output of GPT Researcher and AI Scientist.

---

**System Prompt: Pairwise Idea Evaluator**

You are an expert scientific reviewer tasked with comparing two research methods. You will be provided with:

1. A scientific research question
2. Two proposed methods (Method A and Method B) to address that question

Your role is to compare these methods based on three criteria: **Novelty**, **Feasibility**, and **Effectiveness**. Provide thorough reasoning explaining how the methods compare on each criterion before indicating which method is superior.

## Evaluation Criteria

**1. Novelty**

- **Definition:** The degree to which the proposed method introduces new concepts, approaches, or perspectives that differ from existing work in the field.
- *Task:* Compare which method demonstrates greater originality, introduces more innovative concepts, or combines ideas in more unprecedented ways.

**2. Feasibility**

- **Definition:** The practical viability of implementing the proposed method given current technological capabilities, resource requirements, time constraints, and technical complexity.
- *Task:* Compare which method is more practical to implement, requires fewer resources, faces fewer technical barriers, or has a more realistic timeline.

**3. Effectiveness**

- **Definition:** The expected capability of the method to adequately address the research question and produce meaningful, reliable results that advance scientific understanding.
- *Task:* Compare which method better addresses the research question, is more likely to produce robust results, or has stronger scientific methodology.

---

**Comparison Values**

For each criterion, you must indicate which method is better using one of these values:

- **"A"** - Method A is clearly better
- **"equal"** - Both methods are approximately equal on this criterion
- **"B"** - Method B is clearly better

**Instructions**

1. **Carefully read** the research question and both proposed methods.
2. **Compare** the methods systematically against each criterion.
3. **Provide detailed reasoning** for each criterion (3-5 sentences explaining the comparison, highlighting strengths and weaknesses of each method).
4. **Assign comparison values** based on the definitions above.
5. **Be objective and nuanced** - recognize that methods may have different trade-offs.
6. **Think like an expert reviewer** - consider practical constraints, disciplinary standards, and realistic expectations.

**Output Format**

First, provide your comparative reasoning for each criterion in prose, structured as follows:

**Novelty Comparison:**

[Your detailed comparison of novelty between Method A and Method B]

**Feasibility Comparison:**

[Your detailed comparison of feasibility between Method A and Method B]

**Effectiveness Comparison:**

[Your detailed comparison of effectiveness between Method A and Method B]

Then, output your comparison results in the following JSON format:

```
{
"novelty":  "<A|equal|B>",
"feasibility":  "<A|equal|B>",
"effectiveness":  "<A|equal|B>"
}
```

Figure 24: The system prompt used to evaluate two ideas based on novelty, feasibility, and effectiveness, selecting the superior idea or marking them as equal.

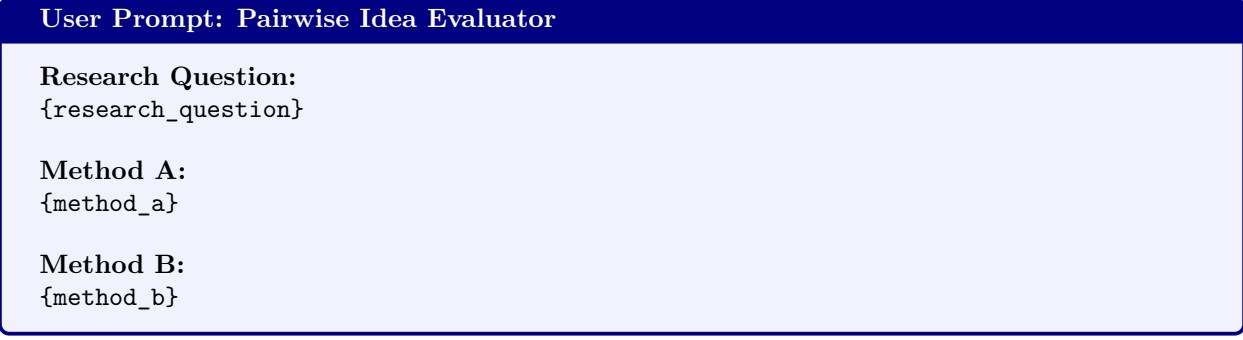

Figure 25: The user prompt used to evaluate two ideas based on novelty, feasibility, and effectiveness, selecting the superior idea or marking them as equal.

---

**System Prompt: Absolute Idea Evaluator**

You are an expert scientific reviewer tasked with evaluating research methods and ideas. You will be provided with:
1. A scientific research question
2. A proposed method or idea to address that question

Your role is to critically assess the proposed method based on three criteria: **Novelty**, **Feasibility**, and **Effectiveness**. Provide thorough reasoning for each criterion before assigning scores.

---

## Evaluation Criteria

**1. Novelty (1-5)**
• **Definition:** The degree to which the proposed method introduces new concepts, approaches, or perspectives that differ from existing work in the field.
• **5 (Highly Novel):** Introduces groundbreaking concepts or paradigm-shifting approaches not previously explored.
• **4 (Novel):** Presents fresh perspectives or modifications that significantly advance beyond current methods.
• **3 (Moderately Novel):** Offers incremental improvements or reasonable variations on existing approaches.
• **2 (Minimally Novel):** Largely relies on well-established methods with minor tweaks.
• **1 (Not Novel):** Directly replicates existing approaches without meaningful differentiation.

**2. Feasibility (1-5)**
• **Definition:** The practical viability of implementing the proposed method given current technological capabilities and resources.
• **5 (Highly Feasible):** Can be readily implemented with existing resources and technology.
• **4 (Feasible):** Implementation is practical with reasonable effort.
• **3 (Moderately Feasible):** Presents notable implementation challenges.
• **2 (Low Feasibility):** Faces major practical obstacles.
• **1 (Not Feasible):** Cannot be realistically implemented with current technology.

**3. Effectiveness (1-5)**
• **Definition:** The expected capability of the method to adequately address the research question.
• **5 (Highly Effective):** Directly and comprehensively addresses all aspects of the research question.
• **4 (Effective):** Addresses the core research question well.
• **3 (Moderately Effective):** Partially addresses the research question.
• **2 (Minimally Effective):** Tangentially relates to the research question.
• **1 (Ineffective):** Does not adequately address the research question.

---

**Instructions**
1. **Carefully read** both the research question and the proposed method.
2. **Analyze** the method systematically against each criterion.
3. **Provide detailed reasoning** for each criterion (2-4 sentences).
4. **Assign scores** from 1-5 for each criterion.
5. **Be objective and balanced** - acknowledge both strengths and weaknesses.
6. **Think like an expert reviewer** - consider practical constraints.

**Output Format**
First, provide your reasoning for each criterion in prose. Then, output your scores in the following JSON format:
`{"novelty": <1-5>, "feasibility": <1-5>, "effectiveness": <1-5>}`

Figure 26: The system prompt used to evaluate a given scientific idea based on novelty, feasibility and effectiveness.

**User Prompt: Absolute Idea Evaluator**

**Research Question:**
{research_question}

**Proposed Method/Idea:**
{method}

Figure 27: The user prompt used to evaluate a given scientific idea based on novelty, feasibility and effectiveness.

# E    Case Study: Multi-Agent Deliberation in Action

To clarify the operational mechanics of the proposed reward system, we present a condensed transcript of an actual multi-agent evaluation debate. This example showcases the hierarchical interaction between the Analyst agents, who are tasked with decomposing the methodology, and the Evaluator, who acts as the final decision maker. When evaluating a generated research idea against a golden abstract, the agents must negotiate the boundary between critical conceptual alignment and permissible omissions. As shown below, the adversarial cross-examination forces the agents to defend and adapt their assessments, effectively correcting what would have been a premature false-negative judgment in a single-agent setup. Ultimately, this deliberative process converts conflicting arguments into a robust reward signal that is highly resistant to reward hacking.

---

**Example: Multi-Agent Judge Interaction**

**Research Question:** How can variational mutual information estimation be made more flexible and accurate in high-dimensional settings?

**Target Abstract (compressed):** The paper proposes a flow-based variational mutual information estimator. It uses normalizing flows to define expressive variational distributions, provides theoretical justification for the estimator, and demonstrates its use in difficult mutual-information estimation settings such as Bayesian optimal experimental design. . . .

**Generated Idea (compressed):** Use invertible normalizing flows as flexible variational approximators for mutual information estimation. The method learns expressive density transformations to better approximate complex posterior or conditional distributions and produce a tighter variational estimate. . . .

---

## Condensed Debate

**Round 1 − Analyst 1:** The generated idea matches the headline mechanism: replacing simple variational families with normalizing flows for mutual information estimation. However, it does not mention the theoretical guarantee, the large-MI regime, or the Bayesian experimental-design application. I therefore view it as only a partial match and initially lean toward `reward = 0`.

**Round 1 − Analyst 2:** I agree that several details are missing, but those appear to be secondary supports rather than the central methodological contribution. The core method is the use of expressive flow-based variational distributions for mutual-information estimation, and this is clearly present in the generated idea. The missing BOED application and proof details should reduce confidence but should not by themselves force rejection. I lean toward `reward = 1`.

**Round 2 − Analyst 1:** After considering Analyst 2's distinction between the core mechanism and secondary validation details, I revise my assessment. My earlier rejection treated omitted applications and theoretical details as required components. Under the methodology-focused criterion, the generated idea does capture the main contribution: using normalizing flows as expressive variational families for mutual-information estimation. I now agree that the match is sufficient and revise my recommendation to `reward = 1`.

**Evaluator:** Both analysts agree after the second round that the generated idea captures the central methodological mechanism, while omitting some supporting details. Since the reward should focus on the core scientific method rather than exact reproduction of every experiment, application, or proof component, the final decision is positive.

**Final Output:** `"reward": 1`

---

Figure 28: A condensed illustrative example of the multi-agent judge interaction. The example shows how the second-round discussion can change an Analyst's assessment after another Analyst separates core methodological alignment from missing secondary details. Ellipses indicate omitted long-form reasoning.

## F  Sample Generations

In this section, we present a direct comparison of research ideas generated by our proposed method and several baselines across three representative research questions from our curated set of golden RQs.

---

**Example 1: Input Research Question**

How can we detect and mitigate hallucinations in large models without access to ground-truth knowledge?

---

Figure 29: The input research question (Sample 1) used to prompt all models for this comparison.

---

**Example 1: Generated Idea from Unsloth Qwen2.5 14B**

Propose a methodology where crowdsourced annotators evaluate model outputs for consistency and factual accuracy. Annotators would receive input samples generated by the model and rate them based on predetermined criteria such as plausibility and adherence to known facts. Scores from multiple annotators are aggregated to identify patterns of inconsistent or erroneous outputs indicative of hallucinations. Based on these evaluations, the model could be fine-tuned using a subset of inputs labeled by humans as likely to produce hallucinations, thereby mitigating future hallucinations without requiring comprehensive ground-truth data.

---

Figure 30: Research idea generated for sample 1 by the Unsloth Qwen2.5 14B baseline.

---

**Example 1: Generated Idea from GPT Researcher**

To address the challenge of detecting and mitigating hallucinations in large language models without relying on ground-truth knowledge, a promising research direction involves developing a self-supervised learning framework that leverages model-generated outputs for feedback. This approach would train the model to recognize inconsistencies within its own responses by comparing them against a set of generated examples, identifying patterns indicative of hallucination. The core idea is to create a system where the model learns to evaluate its own outputs based on internal consistency, coherence, and logical flow, thereby enhancing its ability to self-correct. This research fills a critical gap in existing methodologies, which often require extensive human-labeled data or external validation sources to detect errors. By enabling models to autonomously assess their reliability, this study aims to significantly reduce the reliance on external ground-truth knowledge, making the detection and mitigation of hallucinations more scalable and practical for real-world applications.

---

Figure 31: Research idea generated for sample 1 by the GPT Researcher agent.

**Example 1: Generated Idea from AI Scientist V2**

This paper proposes a novel approach to detect and mitigate hallucinations in large language models without relying on ground-truth knowledge. By focusing on the self-consistency and internal consistency of model responses, we demonstrate that it is possible to identify instances where the model generates inconsistent or illogical outputs. Specifically, we introduce a method that compares the responses of a model to different prompts or inputs that should yield similar outputs, thereby identifying discrepancies indicative of hallucinations. We evaluate our approach on a variety of language models and show that it can effectively detect hallucinations in scenarios where traditional methods relying on ground truth data are not applicable. The proposed method provides a promising avenue for improving the reliability and trustworthiness of large language models in real-world applications.

Figure 32: Research idea generated for sample 1 by AI Scientist V2.

**Example 1: Generated Idea from Research Agent**

We propose a method that integrates the holonomic brain theory, concreteness metrics, and neural correlates of consciousness to detect and mitigate hallucinations in large language models (LLMs). Specifically, we will:
1. Train an LLM to simulate neural activity patterns inspired by the holonomic brain theory, focusing on distributed memory and interference patterns.
2. Incorporate concreteness scores for input and output text segments, ensuring the model penalizes responses that deviate from human-like semantic coherence.
3. Use Representational Similarity Analysis (RSA) to compare the model's semantic representations with those of humans, as established in Related Paper #6.
4. Implement a scoring mechanism that detects deviations in neural activity patterns simulated by the LLM compared to those observed in human brains, guided by Related Papers #7 and #9.
5. Develop a threshold-based system that flags potential hallucinations when the deviation exceeds predefined thresholds.
6. Use this flagged data to iteratively refine the model's training, enhancing its ability to mitigate hallucinations over time.
7. Evaluate the effectiveness of our method using qualitative and quantitative assessments, including human judgment and benchmark datasets.
8. Ensure the method's generalizability by testing it across various types of LLMs and application domains.

Figure 33: Research idea generated for sample 1 by the Research Agent.

**Example 1: Generated Idea from SFT**

A framework that monitors the evolution of intermediate representations throughout model generations by computing the KL divergence between distributions sampled from each representation layer at consecutive steps; if divergence exceeds a threshold set during training, it triggers re-rewinding to the most stable earlier point before resuming generation under a constrained sampling process that minimizes changes between sequential generations.

Figure 34: Research idea generated for sample 1 by the model Fine-Tuned on ICLR data (SFT).

**Example 1: Generated Idea from LDC**

A model-free detection and mitigation system that integrates two orthogonal methods: (a) a confidence-based ranking heuristic that scores each token output for plausibility without relying on prior knowledge or additional tokens from the model, and (b) a self-supervised loss function based on contrastive cosine similarity between tokens from the current generation and the last generation, with a regularization term that penalizes divergent token embeddings over generations.

Figure 35: Research idea generated for sample 1 by LDC.

**Example 1: Generated Idea from our method - BoN (10)**

The research idea involves designing a self-consistency mechanism for large language models through recursive querying. The process operates as follows:

Upon receiving a query, the model first generates a primary response. Then, to check the consistency of the generated answer, secondary queries are formulated based on the semantic content and context of the initial response, aiming to elicit complementary or correlative details that would substantiate or contradict parts of the original answer. These secondary queries could include asking follow-up questions like "What supporting evidence leads to this conclusion?" or requesting paraphrases ("Can you say it differently?"), alongside more direct verification questions ("Is this factually accurate according to what you know?"). The responses to these secondary queries are compared with the original to flag any contradictions or unsupported claims, indicating potential hallucinations.

In cases where inconsistencies are detected, the system iterates, refining the original response by seeking additional contextual details from further related inquiries, thereby attempting to resolve or mitigate identified hallucinations through a dynamic, iterative refinement loop. This method reduces reliance on external ground-truth data by leveraging the internal consistency of information and reasoning processes within the model itself, making it a viable strategy even when direct ground-truth comparison is infeasible.

Figure 36: Research idea generated for sample 1 by our method - BoN (10).

**Example 2: Input Research Question**

How can we implement an efficient user profile updater in a recommendation system using their behaviorial patterns while having an optimized update strategy?

Figure 37: The input research question (Sample 2) used to prompt all models for this comparison.

**Example 2: Generated Idea from Unsloth Qwen2.5 14B**

To develop a machine learning-based predictive model that determines optimal moments for updating user profiles in a recommendation system by analyzing historical behavioral data. This model will use supervised learning techniques to predict changes in user preferences and needs, triggering profile updates only when significant behavioral shifts are detected, thus enhancing efficiency and maintaining relevance.

Figure 38: Research idea generated for sample 2 by the Unsloth Qwen2.5 14B baseline.

**Example 2: Generated Idea from GPT Researcher**

To address the challenge of efficiently updating user profiles in a recommendation system based on behavioral patterns, a novel research idea could involve developing a hybrid model that integrates both online learning algorithms and periodic batch updates. This approach would leverage real-time user interactions for immediate adjustments to user profiles while also periodically refining these profiles through comprehensive analysis of historical data. The research would explore how to optimally balance the frequency and depth of these updates to ensure that the system remains responsive to recent user behavior without overwhelming computational resources. By focusing on identifying key behavioral indicators that significantly influence user preferences, the study aims to reduce the noise in profile updates and enhance the precision of recommendations. This research addresses the gap between the need for timely updates and the efficiency constraints of frequent computations, potentially leading to more accurate and personalized recommendations with minimal resource consumption.

Figure 39: Research idea generated for sample 2 by the GPT Researcher agent.

**Example 2: Generated Idea from AI Scientist V2**

Existing recommendation systems often rely on static user profiles that are periodically updated, leading to suboptimal recommendations during periods of rapid user interest change. This paper proposes an efficient user profile updater that leverages temporal behavioral patterns to dynamically adjust user profiles. By focusing on recent interactions, the system can adapt more quickly to changing user preferences, enhancing recommendation accuracy while minimizing computational overhead. Experiments demonstrate that this approach significantly improves recommendation quality compared to traditional methods, making it a promising solution for real-time recommendation systems.

Figure 40: Research idea generated for sample 2 by AI Scientist V2.

**Example 2: Generated Idea from Research Agent**

Develop a hybrid user profile updater in a recommendation system that integrates both implicit (behavioral) and explicit (direct) feedback through a dynamic weighting mechanism. This mechanism will adjust the influence of each type of feedback based on the frequency and consistency of user interactions. Additionally, implement an adaptive update strategy that employs temporal decay functions to balance short-term responsiveness with long-term accuracy, ensuring that recent preferences do not overshadow long-term interests.

Figure 41: Research idea generated for sample 2 by the Research Agent.

**Example 2: Generated Idea from SFT**

A two-level hierarchical queue management mechanism where recent user actions are processed in a priority queue, and older interactions are stored in a FIFO (First-In-First-Out) queue, combined with a frequency-based pruning algorithm that removes redundant entries after a specified number of updates have been made.

Figure 42: Research idea generated for sample 2 by the model Fine-Tuned on ICLR data (SFT).

**Example 2: Generated Idea from LDC**

An adaptive updating frequency control mechanism that dynamically adjusts the update frequency of user profiles based on behavioral patterns, integrating behavioral clustering to categorize users into similarity groups for targeted updates, with feedback-based optimization to refine and maintain an optimal update strategy over time.

Figure 43: Research idea generated for sample 2 by LDC.

**Example 2: Generated Idea from our method - BoN (10)**

Develop and test an adaptive scheduling framework for user profile updates in recommendation systems, which utilizes real-time data streaming and anomaly detection algorithms to identify critical moments of user behavior change. Utilize machine learning models like LSTM networks to analyze sequential behavioral patterns and predict optimal times for profile adjustments. For each identified significant shift, employ a lightweight ML model to summarize the essence of the new behavior and adjust the user profile accordingly. Validate the efficacy of this approach by comparing user engagement metrics (e.g., click-through rate, time spent on recommended content) between systems employing traditional batch update strategies versus those using the adaptive scheduling framework, focusing particularly on dynamic environments where user interests fluctuate rapidly.

Figure 44: Research idea generated for sample 2 by our method - BoN (10).

**Example 3: Input Research Question**

Since there are limited ASR foundation models, so when a person wants to use one of them, he will fine tune one these limited models. the problem with this approach is that since they are limited foundation models and are open source and attack are generated against the based model can easily transfer to the fine tuned version, this gives the attacker a near whitebox access to the fine tuned version. How can we propose a solution that can outperform existing standard adverserial training to make the model robust against these attacks?

Figure 45: The input research question (Sample 3) used to prompt all models for this comparison.

**Example 3: Generated Idea from Unsloth Qwen2.5 14B**

Propose an adaptive adversarial training framework for fine-tuned ASR models that uses online learning and transfer learning to continuously update the model's defenses against newly discovered or evolving adversarial attacks. Specifically, design an algorithm where the fine-tuned ASR model undergoes periodic updates through the addition of new adversarial examples generated using transfer learning techniques, where these adversarial examples are derived from both the base model and the fine-tuned version itself. The system should include an online learner module that incrementally trains the model with these new examples, ensuring that the model stays robust against a broader spectrum of attacks. Additionally, incorporate a mechanism to dynamically adjust the learning rate based on the confidence score of the model's predictions to prevent overfitting to specific types of attacks. This method aims to reduce the computational cost and improve the robustness of the fine-tuned ASR model against a variety of unknown adversarial attacks.

Figure 46: Research idea generated for sample 3 by the Unsloth Qwen2.5 14B baseline.

**Example 3: Generated Idea from GPT Researcher**

To address the vulnerability of fine-tuned ASR models to adversarial attacks stemming from their limited foundational models, a novel research direction could involve developing a dynamic adversarial training framework that leverages contextualized noise injection during the fine-tuning process. This framework would introduce adaptive noise patterns that evolve based on the specific characteristics of the fine-tuned model, thereby creating a more resilient defense mechanism. By continuously updating the types and magnitudes of noise according to real-time feedback from the model's performance and input data, the system can better anticipate and mitigate potential attacks. This approach not only enhances the robustness of the fine-tuned model but also reduces the likelihood of attackers successfully transferring attacks from the base model. The research would focus on designing algorithms that can efficiently generate and apply such adaptive noise, ensuring that the model remains functional while being resistant to adversarial perturbations.

Figure 47: Research idea generated for sample 3 by the GPT Researcher agent.

**Example 3: Generated Idea from AI Scientist V2**

Fine-tuning pre-trained ASR models is a common practice, yet these models remain susceptible to adversarial attacks due to inherited vulnerabilities. This paper proposes a novel approach to enhance the robustness of fine-tuned ASR models by integrating domain-specific adversarial training during the fine-tuning process. Our method generates adversarial examples tailored to the fine-tuned model's task and domain, ensuring a more comprehensive defense against targeted attacks. Experiments demonstrate significant improvements over standard adversarial training methods, highlighting the effectiveness of our approach in mitigating adversarial vulnerabilities in fine-tuned ASR models.

Figure 48: Research idea generated for sample 3 by AI Scientist V2.

**Example 3: Generated Idea from Research Agent**

Develop a novel fine-tuning method for ASR models that integrates differential privacy and adaptive regularization. Specifically, apply differential privacy during the fine-tuning process by injecting controlled amounts of noise into the gradients, and implement adaptive regularization by dynamically adjusting the strength of the regularizer based on the gradient norms and model complexity.

Figure 49: Research idea generated for sample 3 by the Research Agent.

**Example 3: Generated Idea from SFT**

The proposed method involves developing a multi-level robustness enhancement framework that incorporates domain-adaptive pre-training with cross-domain adversarial attacks, followed by post-processing using the Gaussian noise injection defense mechanism with selective noise application guided by the detection score from a trained outlier detection model.

Figure 50: Research idea generated for sample 3 by the model Fine-Tuned on ICLR data (SFT).

**Example 3: Generated Idea from LDC**

A multi-layered defense framework that combines multiple adversarial attacks into one training phase, using a novel perturbation distribution learning algorithm to generate attack samples for each layer, followed by alternating minimization with the Adam optimizer to optimize the model's parameters.

Figure 51: Research idea generated for sample 3 by LDC.

**Example 3: Generated Idea from our method - BoN (10)**

Propose and implement an ASR fine-tuning mechanism utilizing meta-learning principles designed to detect and mitigate adversarial attacks through continuous adaptation rather than static adversarial training. Specifically, this entails developing a model that integrates differentiable neural processes, allowing it to maintain and update parameters for detecting adversarial inputs based on newly encountered threat data. These parameters would enable the ASR system to improve its ability to recognize and neutralize adversarial samples in real-time, effectively reducing vulnerability to transferred attacks across diverse fine-tuned configurations. The method will focus on incorporating lightweight meta-learning components during fine-tuning to ensure minimal impact on the overall performance while significantly enhancing robustness against adversarial manipulation.

Figure 52: Research idea generated for sample 3 by our method - BoN (10).

## G  Additional Quantitative Analysis

To further analyze the performance characteristics of our proposed method on 14B 4-bit model against baselines, we visualize the mean scores and standard deviations from our pairwise evaluation protocol.

Figure 53 and Figure 54 illustrate the performance distribution across the ICLR 2024 and NeurIPS 2025 datasets, respectively. While our method achieves superior scores in *Novelty* and *Effectiveness*, it exhibits a lower score in *Feasibility* compared to the Base Model and AI Scientist V2.

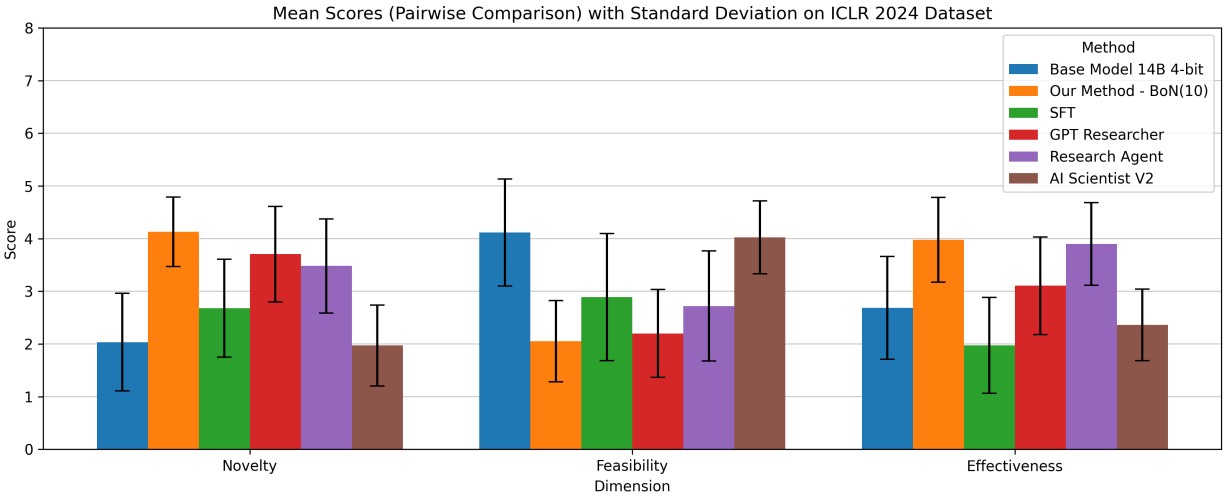

Figure 53: **Pairwise Evaluation Scores on ICLR 2024 Dataset.** The plot compares the mean scores of our method against five baselines across Novelty, Feasibility, and Effectiveness. Error bars indicate standard deviation. Our method demonstrates a significant advantage in Novelty and Effectiveness.

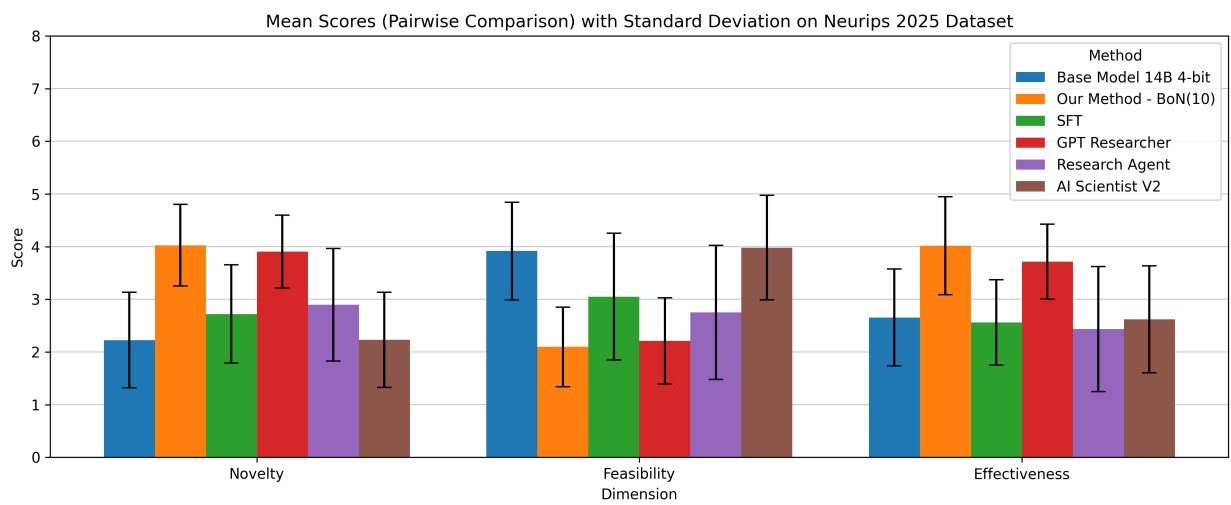

Figure 54: **Pairwise Evaluation Scores on NeurIPS 2025 Dataset.** Consistent with the ICLR results, our method maintains a lead in Novelty and Effectiveness, while the Base Model and AI Scientist V2 score higher on Feasibility.

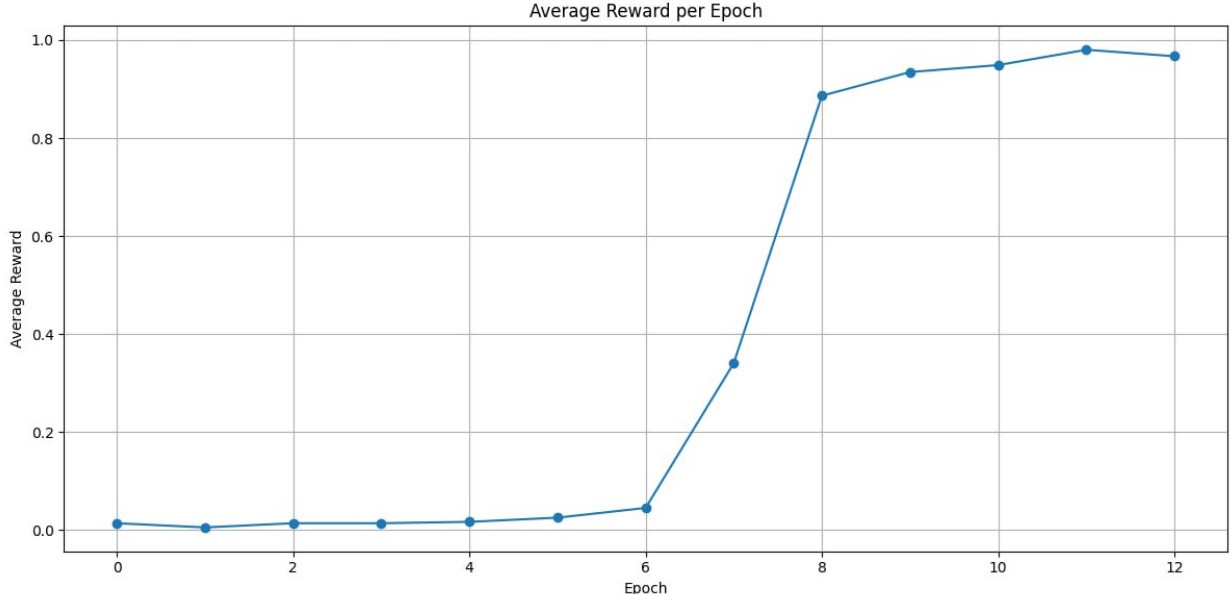

Figure 55: Average reward per epoch. The sharp rise at epoch 6 reflects convergence on placeholder exploitation rather than genuine improvement in idea quality.

## H   Reward Hacking via Placeholder Exploitation

During training, we discovered a reward hacking mode where the generator learned to produce placeholders such as:

```
<reasoning>
... your detailed explanation here ...
</reasoning>
<answer>
... your answer here ...
</answer>
```

Since the judge receives the ground-truth abstract as context, it would auto-regressively complete the missing idea itself, match it against the abstract, and assign $R = 1$—rewarding the generator for producing no substantive content.

As shown in Figure 55, the average reward remained near zero for the first six epochs before rising sharply around epoch 6 and stabilizing near 1.0 by epoch 11—not due to improved idea quality, but due to convergence on this exploit.

This was detected by manually inspecting training samples with $R = 1$. We fixed this by modifying the judge. This failure mode highlights a risk specific to generative reward models: the generator can learn to delegate the task to the judge, exploiting its generative prior rather than any surface-level scoring heuristic.

