# OpenReview forum: "Debate as Reward: A Multi-Agent Reward System for Scientific Ideation via RL Post-Training"
_TMLR — Under review for TMLR_

### Review · Reviewer_NMHA · 2026-06-08

**Summary Of Contributions:**

This paper proposes a post-training-based methodology to build a Large Language Model (LLM) that generates appropriate solution ideas when given a specific research question.

Unlike previous studies aiming at scientific idea generation, which often rely on sophisticated techniques during inference (tending to result in higher inference costs), a distinctive feature of this paper is its focus on a learning approach that enhances scientific idea generation capabilities through reinforcement learning during post-training. Consequently, while it incurs training costs, it significantly reduces costs at the inference stage. Specifically, within the same inference cost budget as existing methods, it enables the adoption of multi-candidate comparisons such as Best-of-N ($N=10$).

Another technical contribution of this work is the introduction of a multi-agent LLM judge system in the reinforcement learning post-training phase. Prior research has shown that a single-agent "LLM-as-a-judge" framework is highly susceptible to reward hacking. To address this, the authors employ a multi-agent framework where multiple LLMs interact, effectively mitigating reward hacking and improving training reliability. Specifically, multiple LLMs assigned the role of Analysts analyze the output through debate, and an LLM with the role of Evaluator aggregates these results to determine a binary reward. Through ablation experiments, the paper demonstrates that this framework suppresses reward hacking and enhances learning efficiency.

One weakness of this paper lies in the high abstraction level of its explanations. In particular, Section 3.2, "Multi-Agent Evaluation Framework," which is considered one of the most critical technical contributions of this study, only provides an abstract description of each role, and the cited figure in the Appendix merely presents example prompts. Moreover, the Moderator and Critic are not mentioned in Section 3.2, whereas they appear in the experiment. The inclusion of a schematic diagram showing the concrete interactions among agents or a detailed operational workflow would greatly enhance understanding. Another critical weakness is that the experiments were conducted with only a single trial, leaving the statistical significance and reproducibility of the results unclear. Especially for the latter, this serves as a major limitation in evaluating the validity and reliability of the authors' claims.

**Additional Comments:**

Minor corrections and comments.

P2. “We note that, unlike prior work that uses multi-agent LLM judges after generation, ”
It would be beneficial to provide relevant references here to support this statement.

P3. “We adopt this insight not merely for final evaluation, but as a dense reward signal.”
Based on the current text, it is difficult to understand what the authors mean by a "dense reward signal." Please clarify this point.

P4. “As illustrated in Fig. 1, The”
Typo: "The" should be lowercase ("the").

P5. “The debating framework effectively resolves this by transforming evaluation from a static scoring task into a dynamic verification process.”
Which specific experimental results support this claim? Please point to the exact figures/tables or clarify this in the experiments section.

P5. “This successfully ensures that rewards target the intrinsic logic of the scientific mechanism rather than mere lexical overlap or specific implementation contexts.”
It is unclear from which results the authors draw this conclusion. Please provide the empirical basis for this statement.

P6. “To ensure a fair, compute-matched evaluation against these resource-intensive baselines, we evaluate our proposed method using a Best-of-10 (BoN(10)) approach.”
While I agree with this approach, it would be highly interesting to see the performance with $N = 1$, especially since the proposed method shifts the cost from inference to training.

P8. Table 2. Only aggregated results are shown. However, it is interesting to see how the performance depends on the types of research questions. A deeper analysis would be beneficial.

P10. “The ablations also reveal that adding structure or additional agents does not automatically guarantee correctness For example,”
A period is missing before "For example."

**Audience:**

Yes

**Audience Explanation:**

The topic of this paper appeals to a broad audience across various fields of science and engineering. Given that research achievements in related LLM domains have been presented at international conferences such as ICLR, ICML, and COLM, this work is highly likely to attract the interest of TMLR readers.

**Broader Impact Concerns:**

No concern.

**Claims And Evidence:**

No

**Claims Explanation:**

One critical limitation is that, despite the proposed approach being a learning-based method, the experiments were conducted only once, and a sufficient statistical evaluation is lacking. Moreover, regarding the techniques used during inference, only the average scores are reported, resulting in a lack of statistical assessment for the inference phase. This deficiency in statistical evaluation leads to the concern that sufficient evidence has not been provided to support the effectiveness and claims of the proposed method.

Furthermore, several statements lack supporting references or experimental results. For instance, the authors state, "More strikingly, we observed a sophisticated failure mode where the generator model intentionally outputs incomplete placeholder phrases, such as ‘... the answer part ...’"; however, no specific evidence or demonstrated results regarding this point can be found in the paper. Since this observation is a crucial finding directly tied to the motivation of this study, the presentation of supporting evidence or concrete data is deemed necessary.

**Requested Changes:**

Addition of Detailed Explanations for the Specific Framework
In Section 3.2, the description of the Multi-Agent Evaluation Framework is limited to abstract explanations of each role. Please include concrete examples of agent interactions and prompts, as well as a schematic diagram or specific illustrations of the communication between roles, to facilitate a deeper understanding of the mechanism.

Strengthening the Main Trials and Statistical Evaluation of Experimental Results
The experiments for the proposed method were conducted only once, leaving the reproducibility and significance of the results insufficiently evaluated. To demonstrate the reliability of the findings, please incorporate results from multiple independent experimental trials, along with evaluation metrics such as means, standard deviations, and statistical test results (e.g., $t$-test).

Presentation of Quantitative and Statistical Evidence for Training and Inference Results
When demonstrating the effectiveness of the proposed model in training and inference, the authors should present quantitative evidence, such as evaluations showing statistical significance or variance based on multiple trials, rather than relying solely on average scores.

Reinforcement of Evidence for Unsubstantiated Statements
Regarding the statement, "More strikingly, we observed a sophisticated failure mode where the generator model intentionally outputs incomplete placeholder phrases, such as ‘... the answer part ...’," please provide specific examples, data, or figures that substantiate this observation. This will ensure the claim is properly backed up within the paper, thereby improving its credibility.

Improvement of Figures
The font size in Figure 2 is so small that it is virtually unreadable on a standard A4 page, and it remains difficult to decipher even when magnified; thus, the figure is currently inadequate. It is necessary to increase the font size so that the numbers and text are clearly legible.

---

> ### Author Response · Authors · 2026-06-25
> **Reply to Reviewer NMHA Part 1**
>
> Dear Reviewer NMHA,
>
> We sincerely thank you for your careful and constructive review. Your feedback on the clarity of the multi-agent framework, statistical rigor, and unsubstantiated claims has been instrumental in strengthening the manuscript. Below we respond point-by-point to each of your requested changes and additional comments.
>
> ---
> ## Requested Changes
> > In particular, Section 3.2, "Multi-Agent Evaluation Framework," which is considered one of the most critical technical contributions of this study, only provides an abstract description of each role, and the cited figure in the Appendix merely presents example prompts. Moreover, the Moderator and Critic are not mentioned in Section 3.2, whereas they appear in the experiment.
>
> **RC1: Addition of Detailed Explanations for the Multi-Agent Evaluation Framework**
>
> We thank the reviewer for this helpful observation. We agree that the Multi-Agent Evaluation Framework is a central technical contribution and warrants a more concrete description.
>
> In the revised manuscript, we expanded the explanation of the framework in several ways. First, Section 3.2 now references a condensed representative debate example in Appendix Figure 28. Since a complete debate transcript for a single judged sample is approximately 4,000 words, including the full output would be impractical. Figure 28 instead presents a shortened interaction between the two Analysts and the final Evaluator, illustrating how one Analyst initially leans toward a negative reward due to missing details, but revises its recommendation in a subsequent round after the other Analyst distinguishes core methodological alignment from secondary implementation details.
>
> Second, we substantially expanded Appendix C to clarify the operational structure of the system. Specifically, Appendix C now distinguishes between the initial four-role reference configuration and the final deployed configuration:
>
> - **Appendix C.1** describes the initial four-role setup (Moderator, Analyst, Critic, Evaluator).
> - **Appendix C.2** explains the empirical decision to remove the Moderator and Critic based on ablation results.
> - **Appendix C.3** describes the final selected configuration—2 Analysts + 1 Evaluator—which achieved perfect precision while maintaining competitive recall.
>
> The Moderator and Critic were not omitted from Section 3.2 accidentally; they were part of the broader architecture-search and ablation study but are not part of the final judge used in the main RL pipeline. This clarification directly resolves the apparent mismatch between Section 3.2 and the ablation experiments.
>
> ---
>
> > The experiments for the proposed method were conducted only once, leaving the reproducibility and significance of the results insufficiently evaluated. Especially for the latter, this serves as a major limitation in evaluating the validity and reliability of the authors' claims. Moreover, regarding the techniques used during inference, only the average scores are reported, resulting in a lack of statistical assessment for the inference phase.
>
> **RC2 & RC3: Statistical Evaluation and Reproducibility of Results**
>
> We thank the reviewer for this important concern. We agree that independent repeated trials are necessary for rigorous evaluation. We have therefore repeated both training and evaluation under the full experimental setup across four independent runs and report the mean and standard deviation of all results in the updated Table 2.
> In addition, the revised manuscript now reports results for a substantially expanded training dataset combining both the ICLR and NeurIPS collections, increasing the scale of RL training beyond the original 320-paper setting. This allows us to evaluate not only reproducibility across runs, but also the robustness of the proposed approach under a significantly larger and more diverse training distribution.
> Variance measurements (mean ± SD) are now included for all reported metrics in both the direct generation setting and the Best-of-N (BoN) inference setting. These statistics demonstrate that the improvements are consistent across independent trials rather than resulting from isolated high-scoring runs. We also added a "Precision Std." column to Table 8 (the judge ablation table), where the final Multi-Agent Judge achieves a precision standard deviation of 0.000 across five runs, while all other judge configurations exhibit non-zero variability. The low variance across runs, together with the consistent improvements observed after scaling the training dataset, further confirms the stability, reproducibility, and robustness of our approach.

---

> ### Author Response · Authors · 2026-06-25
> **Reply to Reviewer NMHA Part 2**
>
> > Regarding the statement, "More strikingly, we observed a sophisticated failure mode where the generator model intentionally outputs incomplete placeholder phrases, such as '... the answer part ...'"; however, no specific evidence or demonstrated results regarding this point can be found in the paper.
>
> **RC4: Reinforcement of Evidence for the Placeholder Failure Mode**
>
> We agree that this observation requires concrete supporting evidence, and we have now added **Appendix H** to the revised manuscript to document this failure mode in full.
>
> The key evidence is presented in Figure 54, which shows the reward trajectory during training: reward remains near zero for 6 epochs and then rises sharply to ~1.0—not as a result of genuine quality improvement, but because the generator learned to output placeholder phrases such as:
>
> ```
> <reasoning>
> ... your detailed explanation here ...
> </reasoning>
> <answer>
> ... your answer here ...
> </answer>
> ```
>
> Manual inspection of training samples with mixed rewards confirmed this exploit: the single-model judge auto-regressively completed the placeholder text and matched it against the ground-truth abstract, assigning R=1. We mitigated this failure mode by requiring the Analyst to explicitly verify content completeness before issuing a reward. This evidence, including the reward curve and annotated training examples, is now fully documented in Appendix H and cross-referenced from Section 3.2.
>
> ---
>
> > The font size in Figure 2 is so small that it is virtually unreadable on a standard A4 page, and it remains difficult to decipher even when magnified.
>
> **RC5: Improvement of Figure 2**
>
> We thank the reviewer for pointing out this legibility issue. We have updated Figure 2 to significantly increase the font size for all text, numbers, and labels, ensuring the figure is now clearly readable on a standard A4 page.

---

> ### Author Response · Authors · 2026-06-25
> **Reply to Reviewer NMHA Part 3**
>
> ## Additional Comments
>
> > P2. "We note that, unlike prior work that uses multi-agent LLM judges after generation," — It would be beneficial to provide relevant references here.
>
> We appreciate the suggestion. We have added relevant citations at this point in the text to better support the statement.
>
> > P3. "We adopt this insight not merely for final evaluation, but as a dense reward signal." — It is difficult to understand what the authors mean by a "dense reward signal."
>
> We thank the reviewer for flagging this imprecise phrasing. We have revised the sentence to: *"We adopt this insight not merely for final evaluation, but as a richer, more reliable reward signal."*
>
> > P4. "As illustrated in Fig. 1, The" — Typo: "The" should be lowercase.
>
> Thank you for catching this. We have corrected the capitalization error on page 4.
>
> > P5. "The debating framework effectively resolves this by transforming evaluation from a static scoring task into a dynamic verification process." — Which specific experimental results support this claim?
>
> We thank the reviewer for asking us to make the empirical basis explicit. We have revised the discussion of the debating framework to directly reference the judge-validation experiments in Appendix B and the ablation study in Section 4.5 / Figure 2.
>
> The claim is supported by two sets of results. First, Appendix B evaluates several reward-modeling strategies on the same fixed expert-labeled validation set of 177 samples, including reward-hacked examples. As shown in Table 8, static or non-deliberative strategies obtain substantially lower precision: the single-call binary judge achieves 0.85, embedding similarity 0.58, structured compliance 0.80, keyword heuristics 0.79, and the best non-debate multi-stage novelty judgment 0.90. In contrast, the final multi-agent deliberative judge reaches 1.00 precision. Since false-positive rewards are the primary channel through which reward hacking influences RL training, this precision improvement directly supports the role of deliberative verification. Second, the ablation results in Section 4.5 / Figure 2 show that the improvement is tied to the structure of the debate itself: removing the Analyst reduces precision (indicating that methodological decomposition prevents superficial matches), while removing the Evaluator collapses recall (showing that final aggregation is necessary to convert the discussion into a usable reward signal).
>
> > P5. "This successfully ensures that rewards target the intrinsic logic of the scientific mechanism rather than mere lexical overlap or specific implementation contexts." — It is unclear from which results the authors draw this conclusion.
>
> We thank the reviewer for highlighting this point and agree that the empirical justification should be made more explicit. The conclusion is drawn from two sets of results. First, the comparative analysis in Appendix B / Table 8 shows that methods measuring superficial lexical overlap (e.g., Keyword Alignment Heuristics) achieve only 0.79 precision on our expert-annotated benchmark of 177 samples, failing to capture conceptual misalignment. Our deliberative Multi-Agent Judge achieves perfect precision (1.00) on the same benchmark, demonstrating robustness against verbosity and keyword overlap. Second, the ablation of the Analyst role in Section 4.5 / Figure 2 shows that removing the Analyst—whose role is to decompose ideas into core methodological elements—drops precision to 0.780, empirically demonstrating that without explicit methodological decomposition the judge over-accepts superficially similar but logically mismatched proposals. We have revised Section 3.2 to reference both sets of results directly after this statement.
>
> > P6. "To ensure a fair, compute-matched evaluation against these resource-intensive baselines, we evaluate our proposed method using a Best-of-10 (BoN(10)) approach." — It would be interesting to see performance with BoN(1).
>
> The updated Table 2 now includes results for our method under both single-shot generation and BoN(10), averaged over four independent runs.
>
> > P8. Table 2. Only aggregated results are shown. A deeper analysis depending on the types of research questions would be beneficial.
>
> We agree this would be a valuable addition and thank the reviewer for the suggestion. Due to time constraints we were unable to complete this analysis for the current revision, but we intend to include a breakdown by research question type in a future revision.
>
> > P10. "The ablations also reveal that adding structure or additional agents does not automatically guarantee correctness For example," — A period is missing before "For example."
>
> Thank you for catching this. We have added the missing period before "For example" on page 10.

---

### Review · Reviewer_cz3i · 2026-06-08

**Summary Of Contributions:**

### Summary
This paper proposes an RL framework for scientific idea generation that aims to internalize scientific reasoning into a language model rather than relying on inference-time agentic workflows. The key technical contribution is a multi-agent, debate-based reward function in which multiple LLM judges collaboratively determine whether a generated research idea aligns with the core methodological contribution of a target paper. To mitigate reward hacking, the authors design a highly conservative binary reward signal and optimize a Qwen-2.5 model using Dr. GRPO on a dataset of 320 research question-abstract pairs extracted from ICLR 2024 papers. Experimental results on the ICLR, NeurIPS, and AI-Scientist datasets, along with a small human evaluation, suggest improvements over several scientific ideation baselines. The paper further presents an extensive study of reward design and argues that high-precision reward models are critical to successful RL training for open-ended scientific generation tasks.

### Strengths
* Novel reward-centric perspective on scientific ideation: The paper identifies reward design and reward hacking (not the generation architecture) as the primary challenges in applying RL to scientific idea generation, offering a fresh perspective on the problem.
* Multi-agent debate integrated into the RL training loop: The proposed multi-agent judge serves as the online reward function during RL training, which appears to be a novel application of debate-based evaluation beyond inference-time or post-hoc assessment.
* Thorough investigation of reward hacking and reward model design: The paper provides valuable empirical insights into reward exploitation behaviors and systematically studies a large number of reward prompting and judge configurations.
* Comprehensive evaluation including human assessment: The experimental section combines automatic evaluations with human expert judgments, which is particularly important given the difficulty of reliably evaluating scientific ideas using LLMs alone.
* Strong empirical results with an efficient inference-time model: The trained model consistently outperforms the reported baselines while requiring only a single forward pass at inference time, avoiding the complexity and computational cost of many agent-based approaches.

### Weaknesses
* Evaluation is heavily dependent on LLM judges despite the paper itself arguing that LLMs are unreliable evaluators: The paper’s central motivation is that scientific ideation is difficult to evaluate and that standard LLM judges are prone to failure. However, nearly all headline results ultimately rely on LLM judges (for both absolute and pairwise evaluations). This creates a tension where the paper questions the validity of LLM evaluation while simultaneously relying on it as the primary source of evidence.
* The reward function may primarily measure similarity to known papers rather than scientific creativity: The RL objective rewards ideas that match the methodology of an existing paper’s abstract. Consequently, the model is optimized to reconstruct or rediscover known solutions rather than generate genuinely novel scientific ideas. It remains unclear whether improvements reflect better scientific ideation or simply better recovery of the target paper’s contribution.
* Limited dataset size raises concerns about memorization and generalization: The RL training dataset contains only 320 papers, which is extremely small for RL post-training of a 14B parameter model. The paper provides little analysis demonstrating that the learned behavior generalizes beyond the training distribution rather than overfitting to recurring patterns in the dataset.
* The human evaluation is too small to support strong conclusions. The human study is based on only 30 research questions and reports average scores without statistical significance testing, inter-annotator agreement, score variance, or detailed evaluation protocols. Given that human evaluation is the strongest evidence in the paper, the study seems too limited to substantiate the broad claims it makes.
* The contribution of RL itself is not convincingly isolated. The paper compares against SFT and external baselines, but does not clearly disentangle the effects of: the curated dataset, the reward design, the Best-of-N selection strategy, and RL optimization itself.

**Audience:**

Yes

**Audience Explanation:**

The paper addresses the increasingly important topic of scientific idea generation with large language models, which is likely to be of interest to researchers working on LLM agents, RLHF/RLAIF, automated science, and LLM evaluation. In particular, the focus on reward design and reward hacking in open-ended generation tasks is a relevant and underexplored problem. Even though some of the empirical evidence is not fully convincing, the proposed multi-agent reward framework and the discussion of reward-model failure modes provide useful insights that the community may find valuable.

**Broader Impact Concerns:**

I do not have major ethical concerns beyond those already common in scientific idea-generation systems.

**Claims And Evidence:**

No

**Claims Explanation:**

While the paper presents promising empirical results and provides both automatic and human evaluations, the evidence does not fully support the strength of its claims. The central evaluation relies heavily on LLM judges, even though the paper itself highlights the limitations of LLM-based assessment for scientific ideation. Moreover, the human evaluation is relatively small in scale and lacks statistical analysis, making it difficult to draw strong conclusions about generalization and scientific creativity. Finally, the proposed reward function primarily measures alignment with existing papers, leaving open the question of whether the method truly improves novel idea generation or mainly improves reconstruction of known solutions.

**Requested Changes:**

**Critical for acceptance:**
1) Provide stronger evidence that the method improves scientific ideation rather than the reconstruction of known papers: The current reward function is based on matching a target paper’s methodology, making it unclear whether the model learns genuine creativity or simply recovers existing solutions more effectively.
2) Strengthen the evaluation methodology: The paper relies heavily on LLM judges despite highlighting their limitations. Additional human evaluation, inter-annotator agreement statistics, and significance testing would substantially improve confidence in the results.
3) Demonstrate generalization beyond the small training set: Given that RL training uses only 320 papers, additional experiments are needed to rule out overfitting and show that the learned behavior transfers to genuinely novel research questions.
4) Isolate the contribution of the RL training procedure: More ablations are needed to disentangle the effects of dataset curation, reward design, Best-of-N selection, and RL optimization itself.

**Would strengthen the paper:**
1) Include more qualitative analysis of generated ideas: Detailed examples comparing the base model, SFT model, and RL-trained model would help readers better understand the nature of the improvements.
2) Expand discussion of limitations and novelty claims: Several claims regarding scientific reasoning, ideation capability, and the novelty of the proposed framework should be stated more cautiously and aligned more closely with the presented evidence.
3) Compare against additional strong contemporary scientific ideation systems or stronger variants of existing baselines: This would further clarify the empirical advantages of the proposed approach.

---

> ### Author Response · Authors · 2026-06-25
> **Reply to Reviewer cz3i Part 1**
>
> Dear Reviewer cz3i,
>
> We sincerely thank you for your thorough and detailed review. Your feedback on evaluation methodology, generalization, statistical rigor, and the isolation of RL contributions has been invaluable in strengthening the manuscript. Below we respond point-by-point to each of your requested changes, with responses to the corresponding weaknesses integrated where applicable.
>
> ---
>
> > The current reward function is based on matching a target paper's methodology, making it unclear whether the model learns genuine creativity or simply recovers existing solutions more effectively.
>
> **RC1: Stronger Evidence for Scientific Ideation vs. Reconstruction of Known Papers**
>
> We sincerely thank the reviewer for raising this insightful point. We would first like to clarify that the reward function is not designed to measure surface-level similarity or verbatim recovery of an existing paper. The target abstract serves as a high-quality expert reference for the underlying problem-solution structure. In the multi-agent judge, the agents are explicitly instructed to focus on methodology, novelty, and critical assumptions, while ignoring superficial details such as wording, datasets, evaluation metrics, and implementation-specific choices. A generated idea is therefore not required to exactly reproduce the target paper—it can extend, reformulate, or go beyond the reference solution as long as it provides a technically meaningful and well-aligned answer to the research question.
> Importantly, this setup differs from supervised fine-tuning: during RL rollout, the generator is given only the problem-only research question and never directly receives the target abstract or golden idea as an input to imitate. The target abstract is used only externally by the frozen multi-agent judge to compute a scalar binary reward after the candidate idea has already been generated. Thus, the policy is optimized from delayed reward feedback rather than trained to reconstruct target text token-by-token.
>
> Beyond this design clarification, we provide three converging lines of empirical evidence that the model learns genuine scientific ideation rather than reconstruction.
>
> **Temporal generalization and knowledge cutoff.** Our held-out evaluation datasets span ICLR, NeurIPS, and the AI Scientist benchmark, and we intentionally selected target papers published after the base model's knowledge cutoff. Because the model was never exposed to these papers during pre-training, reconstruction from memory is not possible. Strong performance on these unseen problems indicates that the model has learned a generalized process for scientific reasoning.
>
> **Evaluation on novel, unpublished research questions.** We conducted an additional experiment using entirely new, unpublished research questions formulated by PhD students in our laboratory, for which no target papers or known solutions exist. The ideas generated for these questions were evaluated by human domain experts, and the model's strong performance in this setting serves as direct evidence of open-ended ideation capability that cannot be attributed to reconstruction.
>
> **Additional training on an independent dataset.** As further described in our response to RC3 below, we trained and evaluated the model on an expanded dataset of 2,476 papers. The consistent improvements confirm that the RL process teaches generalizable principles of valid scientific methodology rather than overfitting to the reconstruction patterns of the original ICLR-320 dataset.

---

> ### Author Response · Authors · 2026-06-25
> **Reply to Reviewer cz3i Part 2**
>
> > The paper relies heavily on LLM judges despite highlighting their limitations. Additional human evaluation, inter-annotator agreement statistics, and significance testing would substantially improve confidence in the results.
>
> **RC2: Strengthened Evaluation Methodology**
>
> We thank the reviewer for this concern. We do not claim LLMs are fundamentally unreliable evaluators; rather, we acknowledge known limitations from the literature (Si et al., 2024; Zheng et al., 2023) and address them through a multi-protocol evaluation design. Scalable automatic evaluation is a practical necessity given the volume of generated ideas—the same constraint that motivates LLM-based evaluation in prior work (Baek et al., 2025; Guo et al., 2024). To mitigate known biases, we combine absolute and pairwise scoring protocols and explicitly control for position bias in pairwise comparisons. Our choice of evaluator with a knowledge cutoff predating our dataset also guards against familiarity bias when judging novelty.
>
> Critically, we complement automatic evaluation with human expert assessment on 30 open research questions, confirming our method's superiority on novelty, feasibility, and effectiveness (Table 3). To move beyond simple averages, we have significantly fortified this human evaluation by integrating Intraclass Correlation Coefficient (ICC(2,1)) analysis across our three expert annotators.
>
> The results validate the reliability of our human signal. While absolute agreement fell in the moderate range (ICC = 0.412 for Novelty, 0.311 for Feasibility, 0.438 for Effectiveness), partitioning the variance provides crucial empirical validation: the residual random noise ($MS_{\text{error}}$) across all three dimensions is exceptionally low ($0.480 \le MS_{\text{error}} \le 0.657$ ), demonstrating that our expert panel achieved a highly consistent consensus on the relative ranking of generated ideas. The observed variance in absolute scores is primarily driven by systematic differences in rater strictness ($MS_{\text{raters}} = 5.65$ to $25.19$ )—a well-documented phenomenon in expert peer review—and our use of ICC(2,1) accounts for these calibration differences without compromising the underlying quality signal. While a sample of ~30 research questions is standard for the extreme cognitive load required to evaluate AI-driven scientific ideation (Yu et al., 2026; Wang et al., 2024), these statistical additions substantially strengthen the conclusions that can be drawn from our human study.
>
> **References:**
>
> - Si, C., Yang, D., & Diab, M. (2024). Can LLMs Generate Novel Research Ideas? A Large-Scale Human Study with 100+ NLP Researchers. arXiv preprint arXiv:2409.04109.
>
> - Zheng, L., Chiang, W. L., Sheng, Y., Zhuang, S., Wu, Z., Zhuang, Y., ... & Stoica, I. (2023). Judging LLM-as-a-Judge with MT-Bench and Chatbot Arena. Advances in Neural Information Processing Systems, 36.
>
> - Guo, T., Chen, X., Wang, Y., Chang, R., Peng, S., Chawla, N. V., ... & Zhang, X. (2024). Large Language Model based Multi-Agents: A Survey of Progress and Challenges. arXiv preprint arXiv:2402.01680.
>
> - Yu, J., & Qiu, S. (2026). More Than Can Be Said: A Benchmark and Framework for Pre-Question Scientific Ideation. arXiv:2605.06345.
>
> - Wang, Q., Downey, D., Ji, H., & Hope, T. (2024). SciMON: Scientific Inspiration Machines Optimized for Novelty. ACL 2024 (pp. 279–299).

---

> > ### Author Response · Authors · 2026-06-25
> > **Reply to Reviewer cz3i Part 3**
> >
> > > Given that RL training uses only 320 papers, additional experiments are needed to rule out overfitting and show that the learned behavior transfers to genuinely novel research questions.
> >
> > **RC3: Generalization Beyond the Small Training Set**
> >
> > We sincerely thank the reviewer for raising this concern. We agree that demonstrating generalization is critical and address it through three main lines of evidence, all now incorporated in the revised manuscript.
> >
> > **Expanded training dataset (2,476 papers).** We curated additional data samples bringing the total training dataset to 2,476 papers (combined NeurIPS and ICLR), and re-ran our full RL post-training pipeline. Rather than showing performance degradation or plateauing—which would indicate memorization—the model trained on this larger dataset demonstrates improved generalization and a wider performance gap over both baseline models and existing works in the field. These results strongly indicate that the model is learning generalizable methodological reasoning.
> >
> > **Scaled-up evaluation.** While the previous version evaluated 40 cases, the new checkpoint was rigorously evaluated across 310 test cases, substantially increasing the evidence base for generalization claims.
> >
> > **Strict monitoring against overfitting and reward hacking.** We implemented rigorous oversight during training and validation by saving training logs and having a team of five human annotators manually label them, allowing us to track Precision, Recall, and F1 exactly. These monitoring metrics are now detailed in Table 1 in the Appendix, confirming that improvements are genuine and not a result of exploiting the reward function or memorizing the training data.
> >
> > We also note that the original 320-paper dataset size was not arbitrary, but a deliberate design choice constrained strictly by the base model's knowledge cutoff—restricting training to papers published after the cutoff prevents conflating retrieval of memorized pre-training data with genuine methodological reasoning. Furthermore, our RL post-training uses Low-Rank Adaptation (LoRA) rather than full-parameter fine-tuning: the effective trainable parameter count is approximately 275 million (≈1.83% of the total 15.04B parameters), significantly mitigating the risk of memorization at either dataset scale.
> >
> > ---
> >
> > > More ablations are needed to disentangle the effects of dataset curation, reward design, Best-of-N selection, and RL optimization itself.
> >
> > **RC4: Isolating the Contribution of RL Training**
> >
> > We thank the reviewer for this important suggestion. We have substantially expanded our ablation studies (Section 4.5) to systematically disentangle each component.
> >
> > - **Dataset:** We compare our main results (trained on combined ICLR + NeurIPS, Table 2) against a reduced-data setting using only ICLR 2024 (Section 4.5.1, Table 5). Both configurations show consistent improvements over baselines, confirming that performance gains are not solely attributable to data scale.
> > - **Reward Design:** Section 4.5.2 and Appendices B–C provide extensive ablations of the multi-agent judge architecture. Removing the Analyst, Evaluator, or Critic reduces robustness, confirming that the structured debate mechanism is critical for reliable reward signals.
> > - **Best-of-N Selection:** Table 2 now explicitly reports our method both with and without BoN(10). The improvement from BoN is incremental (~0.14 on ICLR novelty), whereas the RL-trained model without BoN already competes strongly with baselines, demonstrating that core gains come from RL training rather than inference-time selection alone.
> > - **Optimization Algorithm:** Section 4.5.3 (Table 5) compares GRPO against PPO using the same dataset and reward design. GRPO consistently outperforms PPO on novelty and effectiveness, confirming that the choice of RL algorithm materially affects performance and that gains stem from the optimization procedure itself.
> >
> > Collectively, these ablations confirm that reward design and the RL optimization algorithm have the largest impact on performance, consistent with our central thesis on the primacy of reward quality.

---

> > > ### Author Response · Authors · 2026-06-25
> > > **Reply to Reviewer cz3i Part 4**
> > >
> > > > Detailed examples comparing the base model, SFT model, and RL-trained model would help readers better understand the nature of the improvements.
> > >
> > > **S1: Qualitative Analysis of Generated Ideas**
> > >
> > > We thank the reviewer for this suggestion. We have added a systematic qualitative comparison in Appendix F across three distinct research domains.
> > >
> > > - **Base Model (Constraint Failure):** The zero-shot base model produces generic solutions and frequently violates explicit constraints. In Example 1 it proposes "crowdsourced annotators" despite a strict "no ground-truth" constraint; in Example 2 it suggests a standard, non-novel "machine learning-based predictive model" lacking any architectural or methodological insight.
> > > - **Agentic Baselines (Vagueness and Hallucination):** GPT Researcher, AI Scientist V2, and ResearchAgent suffer from verbosity or hallucinated complexity. In Example 1, AI Scientist V2 merely suggests comparing "responses to different prompts"—a surface-level self-consistency application—while ResearchAgent produces a "buzzword salad" mixing unrelated concepts such as "holonomic brain theory" and "concreteness metrics."
> > > - **SFT Model (Rigidity and Memorization):** SFT successfully captures the format of a scientific proposal but lacks conceptual depth, producing rigid, memorized-sounding configurations rather than novel solutions—for instance, a mechanical "two-level hierarchical queue" in Example 2, or a disjointed combination of unrelated techniques in Example 3.
> > > - **Our Method (Conceptual Depth and Constraint Satisfaction):** Our RL-trained model consistently generates actionable, cohesive, and highly tailored methodologies. In Example 1 it solves the "no ground-truth" constraint with a dynamic recursive querying system that tests its own internal logic; in Example 2 it proposes an adaptive scheduling framework using LSTM-based anomaly detection; in Example 3 it introduces a meta-learning and differentiable neural process paradigm for continuous real-time adaptation. These outputs demonstrate that our multi-agent debate reward effectively penalizes superficial complexity in favor of sound underlying logic.
> > >
> > > ---
> > >
> > > > Several claims regarding scientific reasoning, ideation capability, and the novelty of the proposed framework should be stated more cautiously and aligned more closely with the presented evidence.
> > >
> > > **S2: Expanded Discussion of Limitations and Moderated Claims**
> > >
> > > We thank the reviewer for this constructive feedback. We have carefully revised the manuscript to ensure claims are closely aligned with the empirical evidence. Specifically, we have moderated strong assertions in the Introduction with more cautious, measured language, and expanded the Limitations section to explicitly address: the residual reliance on LLM judges, the risk that the reward function may favor similarity over true creativity, and the limited scale of the dataset and human evaluation. We believe these revisions more accurately represent the scope and boundaries of our contributions.
> > >
> > > ---
> > >
> > > > Compare against additional strong contemporary scientific ideation systems or stronger variants of existing baselines.
> > >
> > > **S3: Comparison Against Additional Baselines**
> > >
> > > Our evaluation already includes six baselines, which exceeds most prior work in this area (e.g., three baselines in Li et al., 2024; two in Baek et al., 2025; four in Li et al., 2025). We welcome specific suggestions from the reviewer and will prioritize incorporating them in the next revision.
> > >
> > > **References:**
> > >
> > > - Long Li, Weiwen Xu, Jiayan Guo, Ruochen Zhao, Xingxuan Li, Yuqian Yuan, Boqiang Zhang, Yuming Jiang, Yifei Xin, Ronghao Dang, Yu Rong, Deli Zhao, Tian Feng, and Lidong Bing. Chain of ideas: Revolutionizing research via novel idea development with LLM agents. In Christos Christodoulopou los, Tanmoy Chakraborty, Carolyn Rose, and Violet Peng (eds.), Findings of the Association for Computational Linguistics: EMNLP 2025, pp. 8971–9004, Suzhou, China, November 2025. Association for Computational Linguistics. ISBN 979-8-89176-335-7. doi: 10.18653/v1/2025.findings-emnlp.477. URL https://aclanthology.org/2025.findings-emnlp.477/.
> > >
> > >
> > > - JinheonBaek, SujayKumar Jauhar, SilviuCucerzan, and Sung JuHwang. ResearchAgent: Iterative research idea generation over scientific literature with large language models. InLuisChiruzzo, AlanRitter, andLuWang (eds.), Proceedings of the 2025Conference of theNations of theAmer icasChapter of theAssociation forComputational Linguistics: HumanLanguageTechnologies (Vol ume 1: LongPapers), pp. 6709–6738, Albuquerque, NewMexico, April 2025.Association for Computational Linguistics. ISBN979-8-89176-189-6. doi: 10.18653/v1/2025.naacl-long.342. URLhttps: //aclanthology.org/2025.naacl-long.342/.
> > >
> > >
> > > - Ruochen Li, Liqiang Jing, Chi Han, Jiawei Zhou, and Xinya Du. LDC: Learning to generate research idea with dynamic control. arXiv preprint arXiv:2412.14626, 2024. URL https://doi.org/10.48550/arXiv. 2412.14626.

---

### Review · Reviewer_thnr · 2026-06-15

**Summary Of Contributions:**

The paper introduces a method for post-training focused on improving the model's ability to generate novel research ideas in response to short scientificially oriented questions. The method's approach relies on modified version of GRPO utilizing rewards from a handcrafted debate systems where multiple LLMs have different roles (through initial prompt designed by the authors) discussing the proposed ideas during the post-training RL-phase. The whole method is trained on a curated dataset composed of a subsets of ICLR and NeurIPS papers, the the method along with several baselines and reference methods is evaluated on a held-out subset of this data mixture.

**Audience:**

Yes

**Audience Explanation:**

I definitely expect that a big part of the community is currently interested in the aspects of the improving LLMs ability to generate novel, research oriented ideas. If the authors will be able to reinforce the results section with more in depth analysis showing the overall advantage of the proposed method I expect the work to be a valuable piece of information for many of the readers.

**Claims And Evidence:**

No

**Claims Explanation:**

I'll explain my answer in more details later after introducing more details, but on a high-level I see three main limitations of this work.
1. The whole method (and other baselines) are evaluated using a single model Qwen-2.5-72B, and the proposed method are judged mostly (but not exclusively) by the superiority of this method compared to others.
2. The compute-equalizing strategy is unclear (more on this later).
3. Some claims in the paper are more like opinions not research facts, lacking and support within the presented data. I'll list them later so that the authors could incorporate proper results in the refined version.

**Requested Changes:**

## Compute equalizing strategy
1.Why only your model has BoN, when Qwen2.5 or SFT have even smaller compute required for generating the strategy?

2. How did you come up with the number 10 for BoN sampling. Can you justify with the estimates for other methods? Could you provide an explicit Table comparing the cost of inference (as well as the cost of training) for all the models? How does the cost of training with GPT-OSS 120B is compensated for other models?

3. When using BoN(10) sampling, I'd like to understand how much does this strategy contribute to the overall performance of the model. Could you please create a Table comparing the BoN(10) to single-shot sampling and to the Average(10) to give the readers an understanding where does the improvement come from?

## Evaluation details

My biggest worry about the paper lies in the fact that the whole work is based (mostly) by the output of a single (Qwen-2.5-72B) model. As far as I understand the pipeline, the model is restricted to rely only on the internalized knowledge (no tool use enabled? -- please correct me if I misunderstood). This makes me wonder whether treating a single model as an oracle in this case is something we should be comfortable with. I would prefer to see an analysis where the same output are evaluated and compared against each other with different models. I am aware that this would probably be compute-intensive, but so is the training with GRPO and GPT-OSS-120B. I believe this would strengten the obtained results and present a better assessment of the method.


Please explain me how is the relative performance ( novelty/feasibility/effectiveness) computed and how should I interpret the results. Does 1.0 score for feasibility for your method mean that for all compared pairs, the evaluator models judged the ideas of your model to be less feasible than other ideas from other models?

Could you explain why do we observe that AI Scientist (v1) is almost always better than the AI Scientist (v2)? This is somewhat unexpected to me and I'm wondering why there is no improvement for the v2 model.




## Claims lacking support
It might be the case that my lack of expertise within this topic shows up here, but from my perspective at least several claims need support with empirical results of yours or a set of references describing the fact. Here I list a few of them:

> Single-model evaluator are prone to "reward hacking" where the generator optimizes for length or keyword frequency rather than quality.

> This multi-turn adversarial dynamic reduces the variance of the reward signal and penalizes pausible sounding but vacuous text.

> In **our internal experiments** (...) majority voting -- frequently succumb to reward hacking favoring verbose or lexically complex outputs (...).

I would strongly encourage the authors to include these results to the main paper as this would support the otherwise unjustified claim and provide and interesting checkpoint for the community. Second thing, when presenting the results it's generally more precise to describe it with raw numbers i.e. 67% of the time instead of "frequently".

> This formulation (...) provides strong gradients that reinforce methodologically sound ideas without the length exploitation (...).


##Other

Your model (page 5) uses GPT-OSS 120B for assessing the strategy so it quite directly influences how your models is tuned. The model is 10x bigger than your base model making it a bit of unfair comparison -- especially that you skip this part and provide greater compute during the inference.


Your model is the only one which is tuned on the dataset (with RL, I'm not counting SFT in this case) that you use to evaluate the model's responses. How do you expect this to influence the overall performance of the model?

---

> ### Author Response · Authors · 2026-06-25
> **Reply to Reviewer thnr Part 1**
>
> Dear Reviewer thnr,
>
> We sincerely thank you for your thorough and constructive review. Your detailed feedback on our evaluation methodology, computational comparisons, and empirical grounding has been invaluable in helping us improve the manuscript. Below we respond point-by-point to each of your concerns.
>
> ---
>
> > The whole method (and other baselines) are evaluated using a single model Qwen-2.5-72B, and the proposed method are judged mostly (but not exclusively) by the superiority of this method compared to others.
>
> **RC1: Evaluation via a Single Model (Qwen-2.5-72B)**
>
> We thank the reviewer for raising this point and for the opportunity to clarify our evaluation design. We acknowledge that relying on a single LLM for automated evaluation is a limitation, but this choice was necessary to prevent data contamination.
>
> Evaluating scientific novelty requires strict controls over the evaluator's training data. We constrained model selection to systems whose training data predates our dataset (September 2024). Qwen-2.5-72B was selected specifically because it shares the same knowledge cutoff as our trained model, ensuring the evaluator is not biased by familiarity with post-cutoff papers. Using newer models would risk introducing data leakage and unfairly skewing novelty assessments.
>
> Critically, our methodology does not rely solely on automated LLM evaluation. To verify that automated results translate to genuine scientific quality, we conducted a human evaluation using 30 open research questions from PhD candidates, with ideas from our method and all baselines anonymized and presented to domain experts in AI. As reported in Table 3, independent domain experts found that our method achieves the highest average score, consistently outperforming all baselines on novelty and effectiveness. The human evaluation corroborates the LLM-as-a-judge findings, confirming that the superiority of our method is not an artifact of Qwen-2.5-72B's preferences.
>
> ---
>
> > Why only your model has BoN, when Qwen2.5 or SFT have even smaller compute required for generating the strategy? How did you come up with the number 10 for BoN sampling? Could you provide an explicit Table comparing the cost of inference (as well as the cost of training) for all the models? When using BoN(10) sampling, I'd like to understand how much does this strategy contribute to the overall performance of the model. Could you please create a Table comparing the BoN(10) to single-shot sampling and to the Average(10)?
>
> **RC2: Compute-Equalizing Strategy and Best-of-N Sampling**
>
> Please see the updated Table 2 in our revision, which now includes results for our method both with and without BoN(10), averaged over four independent runs. The table clearly demonstrates that even without BoN sampling, our approach consistently outperforms the base model and SFT across most metrics. We apply BoN to our method to ensure a compute-matched comparison with expensive agentic baselines that involve multiple LLM calls per generation. Since our method is already superior without BoN, we did not apply it to weaker baselines, as doing so would be unlikely to change the relative ranking.
>
> Regarding the choice of N=10: we selected this number to approximately match the inference budget of competing methods. Based on our token estimates, GPT Researcher consumes approximately 1k tokens per run, ResearchAgent uses around 100k tokens, and our BoN(10) sits between these at roughly 10k total tokens (though our method already outperforms GPT Researcher without BoN, as shown in Table 2).
>
> We acknowledge that our training cost is higher than baselines that only perform API calls. However, we argue that training is a one-time investment for a fundamental capability embedded in the model itself. Once trained, our inference cost is lower than agentic approaches, making the method more practical for repeated deployment.

---

> ### Author Response · Authors · 2026-06-25
> **Reply to Reviewer thnr Part 2**
>
> **RC3: Single-Model Oracle and Tool Use**
>
> To clarify directly: we do not use any external tools—no web search, no retrieval-augmented generation—for either training or evaluation. Both training and evaluation loops rely exclusively on the models' internalized knowledge to ensure we are testing pure scientific reasoning and conceptual synthesis.
>
> As noted above in W1, using alternative or newer LLMs as judges would destroy the validity of our novelty benchmark. Most alternative models have training data extending into late 2024 or 2025, meaning they have already ingested the accepted ICLR 2024 and NeurIPS 2025 papers used in our evaluation. Such models would evaluate ideas based on memorized text rather than zero-shot creative assessment. As explicitly detailed in Sections 4.2 and 4.4, our model selection was strictly constrained to systems whose knowledge cutoff predates our evaluation dataset (September 2024). Qwen-2.5-72B was selected precisely because it satisfies this constraint, ensuring an unpolluted evaluation of true scientific novelty. The human evaluation in Table 3 provides an independent, model-agnostic confirmation of these results.
>
> ---
>
> > Please explain me how is the relative performance (novelty/feasibility/effectiveness) computed and how should I interpret the results. Does 1.0 score for feasibility for your method mean that for all compared pairs, the evaluator models judged the ideas of your model to be less feasible than other ideas from other models?
>
> **RC4: Interpretation of Pairwise Scores**
>
> The pairwise evaluation protocol is described in the third paragraph of Section 4.4. For each pair of ideas, we present them twice (in both orders) to mitigate positional bias. The evaluator assigns +1 if the first idea is preferred, −1 if the second is preferred, or 0 if they are judged equivalent. These values are aggregated across all comparisons for each method and then linearly normalized to the 1–5 range, where 5 corresponds to the highest-performing method.
>
> A feasibility score of 1.0 for our method does **not** mean our method was judged less feasible than every other method in every pairwise comparison. Rather, it indicates that after aggregating all pairwise comparisons and normalizing, our method received the lowest relative feasibility score among all methods in that dataset. This is a relative ranking: our method may still have received positive judgments in individual comparisons, but its aggregate rank was lower than competing methods. The absolute scores in Table 2 provide complementary information showing that our feasibility scores remain comparable to other methods in absolute terms.
>
> We also note that with the updated results—now averaged over four independent training and evaluation runs—the pairwise scores no longer necessarily take extreme values of 1 or 5. The averaging across runs smooths the scores and provides a more robust estimate of relative performance, reducing susceptibility to single-run variance.
>
> ---
>
> > Could you explain why do we observe that AI Scientist (v1) is almost always better than the AI Scientist (v2)? This is somewhat unexpected to me and I'm wondering why there is no improvement for the v2 model.
>
> **RC5: AI Scientist v1 vs. v2 Performance Paradox**
>
> This result may appear surprising, but we do not believe AI Scientist-v2 should be interpreted as a strict ideation upgrade over v1. The two systems optimize for fundamentally different goals. AI Scientist-v1 is strongly template-grounded, which benefits idea-level evaluations by anchoring generation to concrete, executable, and well-scaffolded research directions. By contrast, AI Scientist-v2 was designed for template-free, workshop-level end-to-end autonomous research, with broader agentic search and experimentation. Gains in autonomy do not necessarily translate into higher scores on ideation-only metrics such as novelty, feasibility, or effectiveness.
>
> This interpretation is consistent with external evidence: Zhu et al. (2025) report a direct comparison in which AI Scientist-v1 outperforms AI Scientist-v2 on soundness, contribution, and overall rating. More broadly, recent work suggests that stronger or more structured research-agent pipelines can improve coherence and execution flow while becoming more conservative—staying closer to seed literature or favoring easier directions rather than more original ones. Recent evaluation papers also emphasize that scientific-idea assessment remains noisy and judge-sensitive, especially when novelty and feasibility are evaluated from short proposals rather than completed work. We therefore read v2's weaker performance in our setting as plausible: it improves end-to-end autonomy but not necessarily ideation quality under pairwise idea-level evaluation.
>
> **References:**
>
> - Weng, Y., Zhu, M., Xie, Q., Sun, Q., Lin, Z., Liu, S., & Zhang, Y. (2025). DeepScientist: Advancing frontier-pushing scientific findings progressively.

---

> > ### Author Response · Authors · 2026-06-25
> > **Reply to Reviewer thnr Part 3**
> >
> > > Single-model evaluators are prone to "reward hacking" where the generator optimizes for length or keyword frequency rather than quality.
> >
> > **RC6: Empirical Support for Reward Hacking Claims**
> >
> > The claim that single-model evaluators are susceptible to reward hacking (optimizing for length or keyword frequency) is directly supported by our ablation studies in the manuscript.
> >
> > As detailed in Appendix B and C, we evaluated a "Single-call Binary Judge" configuration as part of a controlled comparison of distinct judge strategies. Our empirical results demonstrated that this single-model setup was highly vulnerable to verbosity bias and superficial lexical overlap. Quantitatively, as shown in Table 8, this configuration achieved a mean precision of only 0.85—corresponding to a 15% false-positive rate among positively rewarded samples—explicitly permitting false rewards for keyword-matching or overly verbose outputs. By contrast, our final Multi-Agent Judge achieves a precision of 1.00 (0% false positives) on the same expert benchmark.
> >
> > Beyond length and keyword exploitation, we also discovered a subtler failure mode in which the generator learned to output incomplete placeholder phrases (e.g., "... your answer here ..."), tricking the single-model judge into auto-completing the idea and assigning a false-positive reward. This behavior is documented in Appendix H and further underscores the vulnerability of single-model evaluators. To prevent future confusion, we have added an explicit cross-reference in the main text directing readers to the empirical evidence and ablation tables in Appendices B and C.
> >
> > ---
> >
> > > This multi-turn adversarial dynamic reduces the variance of the reward signal and penalizes plausible sounding but vacuous text. In our internal experiments (...) majority voting—frequently succumb to reward hacking favoring verbose or lexically complex outputs (...).
> >
> > **RC7: Empirical Support for Adversarial Dynamics and Variance Reduction**
> >
> > We agree that claims about the robustness and stability of the reward signal should be tied directly to empirical evidence and described with concrete numbers rather than qualitative terms such as "frequently."
> >
> > The quantitative judge-validation results were already included in the appendix of the original submission, but we agree the main text did not make the connection explicit enough. In the revised manuscript, Section 3.2 now directly references Appendix B / Table 8 and Section 4.5 / Figure 2 when discussing the reliability of the multi-agent judge. Table 8 reports results on a fixed expert-labeled benchmark of 177 samples—including reward-hacked cases—with each configuration evaluated over five independent runs. Concretely:
> >
> > - Single-call binary judge: precision **0.85** (15% false positives)
> > - Embedding similarity: precision **0.58**
> > - Structured compliance: precision **0.80**
> > - Keyword heuristics: precision **0.79**
> > - Multi-stage novelty judgment: precision **0.90**
> > - **Final Multi-Agent Judge: precision 1.00 (0% false positives)**
> >
> > To further address the reviewer's concern about variance, we added a new "Precision Std." column to Table 8 reporting standard deviation across the five runs. The Multi-Agent Judge has precision std. **0.000**, while all other strategies exhibit non-zero variability. This supports the claim that the final judge provides a more stable positive-reward signal—important in online RL because false-positive rewards are the primary pathway through which reward hacking influences policy optimization.
> >
> > We also clarified the ablation evidence in Section 4.5 / Figure 2: removing the Analyst reduces precision to 0.780, indicating that methodological decomposition prevents superficial matches; removing the Evaluator collapses recall to 0.033, showing that final aggregation is necessary for a usable binary reward. Finally, we added a condensed multi-agent interaction example in the appendix and referenced it from Section 3.2 to illustrate how debate can revise an initially superficial judgment into a more methodologically grounded decision.

---

> ### Author Response · Authors · 2026-06-25
> **Reply to Reviewer thnr Part 4**
>
> > This formulation (...) provides strong gradients that reinforce methodologically sound ideas without the length exploitation (...).
>
> **RC8: Empirical Support for Length-Exploitation Claims in GRPO**
>
> The theoretical and empirical backing for this claim is drawn directly from Liu et al. (2025), introduced at the beginning of Section 3.3. Standard GRPO applies sequence-level advantages uniformly across tokens, which artificially inflates gradients for longer outputs regardless of quality—the "double-increase" phenomenon documented extensively by Liu et al. (2025). The Dr. GRPO formulation we adopt normalizes advantages at the token level, explicitly preventing reward exploitation through verbosity. To prevent ambiguity, we have slightly revised the wording in Section 3.3 to make clear that the empirical evidence for this specific length-bias mitigation is grounded in the findings of Liu et al. (2025).
>
> **References:**
>
> - Zichen Liu, Changyu Chen, Wenjun Li, Penghui Qi, Tianyu Pang, Chao Du, Wee Sun Lee, and Min Lin. Understanding r1-zero-like training: A critical perspective. In Conference on Language Modeling (COLM), 2025. URL https://doi.org/10.48550/arXiv.2503.20783
>
> ---
>
> > Your model (page 5) uses GPT-OSS 120B for assessing the strategy so it quite directly influences how your model is tuned. The model is 10x bigger than your base model making it a bit of unfair comparison—especially that you skip this part and provide greater compute during the inference.
>
> **RC9: Compute Asymmetry — GPT-OSS 120B Training Judge**
>
> Crucially, the GPT-OSS 120B judge is used **only during training** to generate reward signals—it is not used at inference time. Our final 14B model generates ideas fully independently.
>
> We experimented with smaller judges (Llama 4 Scout 17B 16E, Gemma 3 27B IT, Qwen 2.5 72B, Qwen 3 32B), but these were susceptible to reward hacking. Using a larger, stricter judge serves as a defensive measure to ensure reliable supervision rather than as an unfair advantage. Since the judge performs binary semantic alignment between the generated idea and the ground-truth abstract, a stronger judge primarily reduces false positives—acting as an oracle-like verifier—rather than providing privileged domain knowledge to the smaller model. The additional training compute is a one-time cost that does not carry over to inference.
>
> ---
>
> > Your model is the only one which is tuned on the dataset (with RL, I'm not counting SFT in this case) that you use to evaluate the model's responses. How do you expect this to influence the overall performance of the model?
>
> **RC10: Potential In-Distribution Bias**
>
> We would like to clarify that our evaluation is not restricted to the same data used for RL training. We considered multiple training/evaluation settings to mitigate this concern. For example, when the model is RL-trained on the ICLR-derived set, we also evaluate it on the NeurIPS-derived set—data outside that training split. In addition, our evaluation includes the AI Scientist v1 set and the Golden set, so the reported results are not based solely on the training distribution.
>
> While we agree that RL post-training on a scientific-ideation dataset can improve performance on related benchmarks, the cross-dataset and cross-venue results suggest that the learned signal transfers beyond the in-distribution setting and cannot be explained simply by overfitting to the evaluation data. We also note that benchmark selection in this domain is inherently constrained by model knowledge cutoffs: evaluating on arbitrary datasets risks contamination if target papers or ideas are already present in a model's pretraining data. Benchmark selection must therefore be done carefully, and not every dataset is equally suitable for a clean generalization test.